# Double Auctions with Two-sided Bandit Feedback

**Soumya Basu**
Google Mountain View
basusoumya@google.com

**Abishek Sankararaman**
AWS
abishek.90@gmail.com

## Abstract

Double Auction enables decentralized transfer of goods between multiple buyers and sellers, thus underpinning functioning of many online marketplaces. Buyers and sellers compete in these markets through bidding, but do not often know their own valuation a-priori. As the allocation and pricing happens through bids, the profitability of participants, hence sustainability of such markets, depends crucially on learning respective valuations through repeated interactions. We initiate the study of Double Auction markets under bandit feedback on both buyers' and sellers' side. We show with confidence bound based bidding, and 'Average Pricing' there is an efficient price discovery among the participants. In particular, the regret on combined valuation of the buyers and the sellers – a.k.a. the social regret – is $O(\log(T)/\Delta)$ in $T$ rounds, where $\Delta$ is the minimum price gap. Moreover, the buyers and sellers exchanging goods attain $O(\sqrt{T})$ regret, individually. The buyers and sellers who do not benefit from exchange in turn only experience $O(\log T/\Delta)$ regret individually in $T$ rounds. We augment our upper bound by showing that $\omega(\sqrt{T})$ individual regret, and $\omega(\log T)$ social regret is unattainable in certain Double Auction markets. Our paper is the first to provide decentralized learning algorithms in a two-sided market where *both sides have uncertain preference* that need to be learned.

## 1   Introduction

Online marketplaces, such as eBay, Craigslist, Task Rabbit, Doordash, Uber, enables allocation of resources between supply and demand side agents at a scale through market mechanisms, and dynamic pricing. In many of these markets, the valuation of the resources are often personalized across agents (both supply and demand side), and remain apriori unknown. The agents learn their own respective valuations through repeated interactions while competing in the marketplace. In turn, the learning influences the outcomes of the market mechanisms. In a recent line of research, this interplay between learning and competition in markets has been studied in multiple systems, such as bipartite matching markets [29, 30, 39, 7], centralized basic auctions [26, 22]. These works follow the 'protocol model' [2], where multiple agents follow a similar protocol/algorithm, while each agent executes her protocol using only her own observations/world-view up to the point of execution.

In this paper, we initiate the study of the decentralized Double Auction market where multiple sellers and buyers, each with their own valuation, trades an indistinguishable good. In each round, the sellers and the buyers present bids for the goods.[1] The auctioneer is then tasked with creating an allocation, and pricing for the goods. All sellers with bids smaller than the price set by the auctioneer sell at that price, whereas all the buyers with higher bids buy at that price. Each buyer and seller, is oblivious to all the prices including her own. Only a buyer, or a seller participating in the market observes

---

[0]Work done when AS was affiliated with UC Berkeley. All opinions are that of the authors and do not necessarily represent that of the employers

[1]In some literature, the bids of the sellers is called 'asks', but we use bids for both sellers and buyers.

37th Conference on Neural Information Processing Systems (NeurIPS 2023).

her own valuation of the good (with added noise). Notably, our work tackles two-sided uncertainty, whereas the previous works mainly focused on one-sided uncertainty in the market.

Double auction is used in e-commerce [43] – including business-to-business and peer-to-peer markets, bandwidth allocation [24, 23], power allocation [31]. We focus on the 'Average Mechanism' for double auction. Average mechanism guarantees that the auctioneer and the auction participants incur no losses in the trade. It also ensures that each commodity is given to the participant that values it the highest, thus maximizing social welfare. Additionally, average mechanism can be implemented through simple transactions. These properties make average mechanism a suitable choice in large social markets, such as energy markets [32], environmental markets [35], cloud-computing markets [38, 32], bidding on wireless spectrum [18]. Our objective is to design a bandit average mechanism for double auction markets when the participants are a-prioiri unaware of their exact valuation.

Under average mechanism (detailed in Section 2.2) first an allocation is found, by maximizing $K$ such that the $K$ highest bidding buyers all bid higher than the $K$ lowest bidding sellers. The price is set as the average of the $K$-th lowest bid among the buyers, and the $K$ highest bid among the sellers for the chosen $K$. We have two-sided uncertainty in the market, as both buyers and sellers do not know their own valuation. The uncertainty in bids manifests in two ways. Firstly, each buyer needs to compete with others by bidding high enough to get allotted so that she can discover her own price. Similarly, the sellers compete by bidding lower for price discovery. The competition-driven increase in buyers' bids, and decrease in the sellers' bids may decrease the utility that a buyer or seller generates. Secondly, as the valuation needs to be estimated, the price set in each round as a function of these estimated valuations (communicated to the auctioneer in the form of bids) remains noisy. This noise in price also decreases the utility. However, when price discovery is slow the noise in price increases. Therefore, the main challenge in decentralized double auction with learning is to strike a balance between the competition-driven increase/decrease of bids, and controlling the noise in price.

## 1.1 Main Contributions

Our main contributions in this paper are as follows.

1. Our paper is the first to provide decentralized learning algorithms in a two-sided market where *both sides have uncertain preference* that need to be learned. Unlike in the setting with one sided uncertainity only, we identify that with *two-sided uncertainty* in double auction markets *optimism* in the face of uncertainty in learning (OFUL) from both sides causes *information flow bottleneck* and thus not a good strategy. We introduce the notion of *domination of information flow* – that increases the chance of trade and price discovery. The sellers bid the lower confidence bound (LCB), and buyers bid the upper confidence bound (UCB) of their respective valuation. By using UCB bids the buyers, and using the LCB bids the sellers decrease their reward and facilitate price discovery. Formally, with the above bids under average mechanism

- We show that the social welfare regret, i.e. the regret in the combined valuation of all the sellers and the buyers in the market is $O(\log(T)/\Delta)$ regret in $T$ rounds with a minimum reward gap $\Delta$. We also show a $\Omega(\log(T)/\Delta)$ lower bound on social-welfare regret and thus our upper bound is order-optimal.
- For individual regret, we show that each of the the sellers and the buyers that do not participate under the true valuations incur $O(\log(T)/\Delta)$ regret, while the optimal participating buyers and sellers incur a $O(\sqrt{T \log(T)})$ regret. Our upper bound holds for heterogeneous confidence widths, making it robust against the choices of the individual agents.

2. We complement the upper bounds by showing price discovery itself is $\Omega(\sqrt{T})$ hard in the minimax sense. Specifically, we consider a relaxed system where *(i)* the price of the good is known to all, and *(ii)* an infinite pool of resource exists, and hence any buyer willing to pay the price gets to buy, and any seller willing to sell at the price does so. We show under this setup, for any buyer or seller, there exists a system where that agent must incur a regret of $\Omega(\sqrt{T})$. Similarly, we establish a $\Omega(\log(T))$ lower bound for the social-welfare regret by showing that the centralized system can be reduced to a combinatorial semi-bandit and using the results of [14].

## 2 System Model

The market consists of $N \geq 1$ buyers and $M \geq 1$ sellers, trading a *single type* of item which are indistinguishable across sellers. This set of $M + N$ market participants, repeatedly participate in the market for $T$ rounds. Each buyer $i \in [N]$ has valuation $B_i \geq 0$, for the item and each seller $j \in [M]$ has valuation $S_i \geq 0$. No participant knows of their valuation apriori and learn it while repeatedly participating in the market over $T$ rounds.

### 2.1 Interaction Protocol

The buyers and sellers interact through an auctioneer who implements a bilateral trade mechanism at each round. At each round $t \geq 1$, every buyer $i \in [N]$ submits bids $b_i(t)$ and seller $j \in [M]$ submits asking price [2] $s_j(t)$ simultaneously. Based on the bids and asking prices in round $t$, the auctioneer outputs *(i)* subsets $\mathcal{P}_b(t) \subseteq [N]$ and $\mathcal{P}_s(t) \subseteq [M]$ of participating buyers and sellers with equal cardinality $K(t) \leq \min(M, N)$, and *(ii)* the trading price $p(t)$ for the participating buyers and sellers in this round. Subsequently, every buyer $i \in [N]$ is *(i)* either part of the trade at time $t$, in which case she gets utility $r_i^{(B)}(t) := B_i + \nu_{b,i}(t) - p(t)$, or *(ii)* is not part of the trade at time $t$ and receives 0 utility along with a signal that she did not participate. Similarly, each seller $j \in [M]$ is either part of the trade and receives utility $r_j^{(S)}(t) := p(t) - S_j - \nu_{s,j}(t)$, or is informed she is not part of the trade and receives 0 utility. The random variables $\nu_{b,i}(t)$ and $\nu_{s,j}(t)$ for all $i \in [N]$, $j \in [M]$ and $t \in [T]$ are i.i.d., 0 mean, 1 sub-Gaussian random variables.[3]

### 2.2 Average price mechanism

Throughout the paper, we assume the auctioneer implements the average price mechanism in every round $t$. Under this mechanism, at each round $t$, the auctioneer orders the bids by the 'natural order', i.e., sorts the buyers bids in descending order and the seller's bids in ascending order. Denote by the sorted bids from the buyer and seller as $b_{i_1}(t) \geq \cdots b_{i_N}(t)$ and the sorted sellers bids by $s_{j_1}(t) \leq \cdots s_{j_M}(t)$. Denote by the index $K(t)$ to be the largest index such that $b_{i_{K(t)}}(t) \geq s_{j_{K(t)}}(t)$. In words, $K(t)$ is the 'break-even index' such that all buyers $i_1, \cdots, i_{K(t)}$ have placed bids offering to buy at a price strictly larger than the asking price submitted by sellers $j_1, \cdots, j_{K(t)}$. The auctioneer then selects the participating buyers $\mathcal{P}_b(t) = \{i_1, \cdots, i_{K(t)}\}$, and participating sellers $\mathcal{P}_s(t) = \{j_1, \cdots, j_{K(t)}\}$. The price is set to $p(t) := \frac{b_{i_{K(t)}} + s_{j_{K(t)}}}{2}$, and thus the name of the mechanism is deemed as the average mechanism.

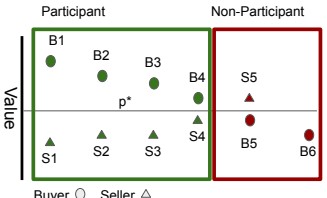

### 2.3 Regret definition

For the given bilateral trade mechanism, and true valuations $(B_i)_{i \in [N]}$ and $(S_j)_{j \in [M]}$, denote by $K^* \leq \min(M, N)$ be the number of matches and by $p^*$ to be the price under the average mechanism when all the buyers and sellers bid their true valuations. Let $\mathcal{P}_b^*$ to be set of the *optimal participating buyers*, and $\mathcal{P}_s^*$ to be set of the *optimal participating sellers*. For any buyer $i \in [N]$, we denote by $(B_i - p^*)$ to be the true utility of the buyer. Similarly, for any seller $j \in [N]$, we denote by

Figure 1: Average Mechanism with 6 Buyers and 5 Sellers

$(p^* - S_j)$ to be the true utility of seller $j$. From the description of the average mechanism, in the system with true valuations, all participating agents have non-negative true utilities.

Recall from the protocol description in Section 2.1 that at any time $t$, if buyer $i \in [N]$ participates, then she receives a mean utility of $(B_i - p(t))$. For a participating seller $j \in [M]$ her mean utility of $(p(t) - S_j)$ in round $t$. If in any round $t$, if a buyer $i \in [N]$ or a seller $j \in [M]$ does not participate, then she receives a deterministic utility 0. The expected individual regret of a buyer $i$, namely $R_{b,i}(T)$,

---

[2]Throughout, we refer to sellers 'bids' as their asking price

[3]We study the system with 1-sub-Gaussian to avoid clutter. Extension to general $\sigma$-sub-Gaussian is trivial when $\sigma$ or an upper bound to it is known.

and a seller $j$, namely $R_{s,j}(T)$, are defined as

$$R_{b,i}(T) = T(B_i - p^*)\mathbb{1}(i \leq K^*) - \mathbb{E}\Big[ \sum_{t:i\in\mathcal{P}_b(t)} (B_i - p(t)) \Big],$$

$$R_{s,j}(T) = T(p^* - S_j)\mathbb{1}(j \leq K^*) - \mathbb{E}\Big[ \sum_{t:j\in\mathcal{P}_s(t)} (p(t) - S_j) \Big].$$

Auctioneer has no regret as average mechanism is *budget balanced*, i.e. auctioneer does not gain or lose any utility during the process.

We also define the social welfare regret similar to gain from trade regret in [10]. The social welfare is defined as the total valuation of the goods after the transfer of goods from seller to buyer in each round. The expected (w.r.t reward noise) total valuation after transfer is thus defined as $\Big( \sum_{i\in\mathcal{P}_b(t)} B_i + \sum_{j\in[M]\setminus\mathcal{P}_s(t)} S_j \Big)$, while the the expected total valuation under oracle average mechanism is $\Big( \sum_{i\in\mathcal{P}_b^*} B_i + \sum_{j\in[M]\setminus\mathcal{P}_s^*} S_j \Big)$. Therefore, the expected social welfare regret is defined as

$$R_{SW}(T) = T\Big( \sum_{i\in\mathcal{P}_b^*} B_i + \sum_{j\in[M]\setminus\mathcal{P}_s^*} S_j \Big) - \mathbb{E}\Big[ \sum_{t=1}^{T} \Big( \sum_{i\in\mathcal{P}_b(t)} B_i + \sum_{j\in[M]\setminus\mathcal{P}_s(t)} S_j \Big) \Big]. \quad (1)$$

# 3 Decentralized Bidding for Domination of Information Flow

We consider the decentralized system where each market participant bids based on their own observation, without any additional communication. The core idea is balancing *domination of information flow* and *over/under bidding*, i.e. ensuring the number of allocation is not less than $K^*$ in each round with high probability, and the bids converge to each agent's true valuation.

Each seller $j \in [M]$, with $n_{s,j}(t)$ participation upto round $t$, at time $t+1$ bids the lower confidence bound (scaled by $\alpha_{s,j}$), LCB($\alpha_{s,j}$) in short, of its own valuation of the item. Each buyer $i \in [N]$, with $n_{s,j}(t)$ participation upto round $t$, at time $t+1$, bids the upper confidence bound (scaled by $\alpha_{b,i}$), UCB($\alpha_{b,i}$) in short, of its own valuation of the item. The bids are specified in Equation 2.

$$s_j(t+1) = \hat{s}_j(t) - \sqrt{\frac{\alpha_{s,j}\log(t)}{n_{s,j}(t)}}, \quad b_j(t+1) = \hat{b}_i(t) + \sqrt{\frac{\alpha_{b,i}\log(t)}{n_{b,i}(t)}} \quad (2)$$

Here, $\hat{s}_j(t) = \frac{1}{n_{s,j}(t)} \sum_{t'\leq t:j\in\mathcal{P}_s(t')} Y_{s,j}(t')$, and $\hat{b}_i(t) = \frac{1}{n_{b,i}(t)} \sum_{t'\leq t:i\in\mathcal{P}_b(t')} Y_{b,i}(t')$ are the observed empirical valuation of the item upto time $t$ by seller $j$, and buyer $i$, respectively.

Our buyers and sellers follow the *protocol model* (which is ubiquitous in bandit learning for markets) and agree on UCB and LCB based bids, respectively. They are *heterogeneous* as they may use different $\alpha_{b,i}$ and $\alpha_{s,j}$ scaling parameters. The only restriction (as seen in Section 4) is $\min\{\alpha_{b,i}, \alpha_{s,j}\} \geq 4$ which they agree on as part of the protocol.

**Key Insights:** We now contrast our algorithm design from standard multi-armed-bandit (MAB) problems. In a typical MAB problems, including other multi-agent settings, algorithm is designed based on optimism in the face of uncertainty in learning (OFUL) principle [1]. The UCB-type indices under OFUL arises as optimistic estimate of the rewards of arms/actions. However, such optimism used from both sides, i.e. both buyer and seller using UCB indices, may lead to a standstill. The bids of the buyers with constant probability can remain below the sellers' bids. Instead, we emphasize information flow through trade. In our algorithm buyers' UCB bids and sellers' LCB bids ensure that the system increases the chance for buyer and seller to participate in each round as compared to using their true valuations. There is *domination of information flow*, and consequently they discover their own valuation in the market.

However, too aggressive over or under bidding can disrupt the price setting process. In particular, if even one non-participating buyer is bidding high enough (due to UCB bids) to exceed a non-participating seller's price, firstly she participates and accrues regret. More importantly, the participating sets deviates, resulting in a deviation of the price of the good from $p^*$. Thus resulting in regret for all participating agents as well. Similar problems arise if one or more non-participating

seller predicts lower. On the other hand, too low aggression is also harmful as the price discovery may not happen resulting in deviation of participating set, deviation of price, and high regret. In the next section, we show the aggression remains within desired range, i.e. the regret of the agents remain low, even with heterogeneous UCBs and LCBs.

# 4 Regret Upper Bound

In this section, we derive the regret upper bound for all the buyers and sellers in the system. Without loss of generality, let us assume that the buyers are sorted in decreasing order of valuation $B_1 > B_2 > ... > B_N$. The sellers are sorted in increasing order of valuation $S_1 < S_2 < ... < S_N$. The the buyers $i = 1, \ldots, K^*$ are the optimal participating buyers, and, similarly, the sellers $j = 1, \ldots, K^*$ are the optimal participating sellers. Let us define $\alpha_{\min} := \min\{\alpha_{b,i}, \alpha_{s,j}\}$, and $\alpha_{\max} := \max\{\alpha_{b,i}, \alpha_{s,j}\}$. We define the minimum distance of an agent's true valuation from the true price $p^*$ as $\Delta = \min_{i,\in[N], j\in[M]}\{|p^* - S_j|, |B_i - p^*|\}$. As an item is sold only if the matched buyer bids strictly greater than the matched seller's bid we have $\Delta > 0$.

Our first main result, Theorem 1, proves a $O(\log(T)/\Delta)$ upper bound for the social welfare regret.

**Theorem 1.** *The expected social welfare regret of the Average mechanism with buyers bidding UCB($\boldsymbol{\alpha}_b$), and sellers bidding LCB($\boldsymbol{\alpha}_s$) of their estimated valuation, for $\alpha_{\min} > 2$, is bounded as:*

$$R_{SW}(T) \le \sum_{i\le K^*} \sum_{i'>K^*} \frac{(\sqrt{\alpha_{\max}}+2)^2}{(B_i - B_{i'})} \log(T) + \sum_{j\le K^*} \sum_{j'>K^*} \frac{(\sqrt{\alpha_{\max}}+2)^2}{(S_{j'} - S_j)} \log(T)$$
$$+ \sum_{j'>K^*} \sum_{i'>K^*} \frac{(\sqrt{\alpha_{\max}}+2)^2}{(S_{j'} - B_{i'})} \log(T) + MNb_{max}\zeta(\alpha_{\min}/2),$$

*where $\zeta(x)$ is the Riemann-zeta function which is finite for $x > 1$.*

The following theorem provides the individual regret bounds for all sellers and buyers in $T$ rounds.

**Theorem 2.** *The expected regret of the Average mechanism with buyers bidding UCB($\boldsymbol{\alpha}_b$), and sellers bidding LCB($\boldsymbol{\alpha}_s$) of their estimated valuation, for $\alpha_{\min} > 2$, is bounded as:*

- *for a participating buyer $i \in [K^*]$ as $R_{b,i}(T) \le (2 + \sqrt{\alpha_{\max}})\sqrt{T\log(T)} + C_{b',i}\log(T)$,*

- *for a participating seller $j \in [K^*]$ as $R_{s,j}(T) \le (2 + \sqrt{\alpha_{\max}})\sqrt{T\log(T)} + C_{s',j}\log(T)$,*

- *for a non-participating buyer $i \ge (K^* + 1)$ as $R_{b,i}(T) \le \frac{\sqrt{(M-K^*+1)}(2+\sqrt{\alpha_{\max}})^2}{(B_{K^*}-B_i)} \log(T)$,*

- *for a non-participating seller $j \ge (K^* + 1)$ as $R_{s,j}(T) \le \frac{\sqrt{(N-K^*+1)}(2+\sqrt{\alpha_{\max}})^2}{(S_j-S_{K^*})} \log(T)$.*

*Here $C_{b',i}$ and $C_{s',j}$ are $O\left(\frac{(M-K^*+1)(N-K^*+1)}{\Delta}\right)$ constants (see Theorem 16 in Appendix A).*

We can summarize our main results (with lower bounds taken from Section 5) as

| | | | Non-participant | |
| --- | --- | --- | --- | --- |
| **Regret** | **Social** | **Participant** | **Buyer** | **Seller** |
| **Upper** | $O(\frac{MN}{\Delta}\log(T))$ | $O((\sqrt{T} + \frac{MN}{\Delta})\log(T))$ | $O(\frac{\sqrt{M}}{\Delta}\log(T))$ | $O(\frac{\sqrt{N}}{\Delta}\log(T))$ |
| **Lower** | $\Omega(\frac{M+N}{\Delta}\log(T))$ | $\Omega(\sqrt{T})^*$ | 0 | 0 |

Table 1: Regret bounds derived in this paper. The participant regret lower bound is instance independent indicated by ($*$). Rest are instance dependent. Lower bound of 0 indicates absence of non-trivial bound.

Several comments on our main results are in order.

**Social and Individual Regret:** The social regret as well as the individual regret of the optimal non-participating buyers and sellers under the average mechanism with confidence bound bids grow as $O(\log(T))$. However, the individual regret of the optimal participating agents grow as $O(\sqrt{T\log(T)})$. The social regret, is determined by how many times a participant buyers fails to participate, and non-participants end up participating. Also, the non-participants incur individual

regret through participation. The effect of price setting is not present in both the cases, and counting the number of bad events lead to $O(\log(T))$ regret. For the participant agents the error in price-setting dominates their respective individual regret of $O(\sqrt{T\log(T)})$.

**Individually rational for $\Delta = \omega(\sqrt{\log(T)/T})$:** We observe that the non-participating buyers and sellers, are only having individual regret $O(\log(T)/\Delta)$. And the participating buyers and sellers are having individual regret $O(\sqrt{T\log(T)})$. This is reassuring as this does not discourage buyers and sellers from participation for a large range of system a-priori. Indeed, for the last participating buyer and seller the utility is $(B_K^* - S_K^*)/2$, which is close to $\Delta$. Hence, as long as $\Delta T = \omega(\max\{\log(T)/\Delta, \sqrt{T\log(T)}\})$ or $\Delta = \omega(\sqrt{\log(T)/T})$ a non-participating buyer or seller prefers entering the market then discovering her price and getting out, as compared to not participating in the beginning. Also, participating buyer or seller is guaranteed return through participation.

**Incentives and Deviations:** We now discuss the incentives of individual users closely following the notion of symmetric equilibrium in double auction [42]. In this setting, a *myopic agent* who knows her true valuation, and greedily maximizes her reward in each round assuming all the non-strategic agents use confidence-based bidding. For average mechanism only the price-setting agents (i.e. the $K^*$-th buyer and the $K^*$-th seller) have incentive to deviate from their true valuation to increase their single-round reward. The *non-price-setting agents are truthful*. When the $K^*$-th buyer deviates then each participating buyer has an average per round surplus of $(B_{K^*} - \max(S_{K^*}, B_{K^*+1}))/2$, and each participating seller has the same deficit. On the contrary, when the $K^*$-th seller deviates each participating seller has average per round surplus of $(S_{K^*} - \min(B_{K^*}, S_{K^*+1}))/2$ surplus, and each participating buyer has the same average deficit in each round. See Appendix C for more discussions.

**Scaling of Regret with $N$, $M$, $K^*$, $\Delta$:**

*Social regret:* The social regret scales as $O((MN - (K^*)^2)\log(T)/\Delta)$. When participant buyers are replaced by non-participant buyers the valuation decreases which leads to $O(K^*(N-K^*)\log(T)/\Delta)$ regret. Similarly, participant seller getting replaced by non-participant seller introduces $O(K^*(M - K^*)\log(T)/\Delta)$ social regret. Finally, as goods move from non-participant buyers to non-participant sellers we obtain $O((M - K^*)(N - K^*)\log(T)/\Delta)$ regret.

*Non-participant regret:* The individual regret scales as $O(\sqrt{M}\log(T)/\Delta)$ for a non-participant buyer, whereas it scales as $O(\sqrt{N}\log(T)/\Delta)$ for a non-participant seller. For any non-participant buyer once it has $O(\log(T)/\Delta^2)$ samples it no longer falls in top $K^*$ buyer. However, it can keep participating until each non-participant seller collects enough samples, i.e. $O(\log(T)/\Delta^2)$ samples. Finally, regret is shown to be $O(\sqrt{\#\text{participation}\log(T)})$ which leads to the $O(\sqrt{M}\log(T)/\Delta)$.

*Participants:* The leading $O(\sqrt{T\log(T)})$ term in the regret for each optimal participating buyer and seller do not scale with the size of the system. This leading term depends mainly on the random fluctuation of the bid of the lowest bidding participating buyer and the highest bidding participating seller. The $O\left(\frac{MN}{\Delta}\log(T)\right)$ regret for the participating buyer and seller comes because each time a non-participating buyer or seller ends up participating the price deviates.

*Special Case $K^* \approx N \approx M$:* We see that when the system the number of participants is very high, i.e. $(N - K^*), (M - K^*) = O(1)$, then the $O(\log(T)/\Delta)$ component, hence the regret, per buyer/seller does not scale with the system size. Furthermore, the social regret also scales as $O((MN - (K^*)^2)\log(T)/\Delta) \approx O(\log(T)/\Delta)$. This indicates that there is rapid learning, as all the participants are discovering her own price almost every round.

## 4.1 Proof Outline of Regret Upper Bounds

We now present an outline to the proof of Theorem 2. The full proof is given in Appendix A. A salient challenge in our proof comes from *two-sided uncertainty*. In Double auction, the outcomes of the buyers' and sellers' sides are inherently coupled. Hence error in buyers' side propagates to the sellers' side, and vice versa, under two-sided uncertainty. For example, if the buyers are bidding lower than their true valuation then the optimal non-participating sellers, even with perfect knowledge of their own valuation, ends up participating. We show that optimal participants (both buyers and sellers) get decoupled, whereas in optimal non-participants regret leads to new information in both buyer and seller side. This breaks the two-sided uncertainty obstacle.

**Monotonicity and Information Flow:** Our proof leverages *monotonicity* of the average mechanism, which means if the bid of each buyer is equal or higher, and simultaneously bid of each seller is equal or lower, then the number of participants increases (Proposition 7). Additionally, UCB ensures each buyer w.h.p. bids higher than its true value, and LCB ensures each seller w.h.p. bids lower than its true value. Therefore, we have *domination of the information flow*: $K(t) \geq K^*$ w.h.p. for all $t \geq 1$.

**Social Regret Decomposition:** For any sample path with $K(t) \geq K^*$ the social regret, namely $r_{SW}(T)$, under average mechanism can be bounded as

$$
r_{SW}(T) \leq \sum_{i \leq K^* < i'} (B_i - B_{i'}) \sum_{t=1}^{T} \mathbb{1}(i \notin \mathcal{P}_b(t), i' \in \mathcal{P}_b(t))
$$

$$
+ \sum_{j \leq K^* < j'} (S_{j'} - S_j) \sum_{t=1}^{T} \mathbb{1}(j \notin \mathcal{P}_s(t), j' \in \mathcal{P}_s(t)) + \sum_{i', j' > K^*} (S_{j'} - B_{i'}) \sum_{t=1}^{T} \mathbb{1}(j' \in \mathcal{P}_s(t), i' \in \mathcal{P}_b(t)).
$$

The first term corresponds to a non-participant buyer replacing a participant buyer. The second term is the same for sellers, whereas the final term corresponds to two non-participant buyer and seller getting matched.

**Individual Regret Decomposition:** We now turn to the individual regrets. Regret of a non-participant buyer $i$, can be bounded by

$$
\sum_{t:i \in \mathcal{P}_b(t)} (p(t) - B_i) \leq \sum_{t:i \in \mathcal{P}_b(t)} (b_i(t) - B_i) \lesssim \sum_{t:i \in \mathcal{P}_b(t)} \sqrt{\frac{\alpha_{b,i} \log(t)}{n_{b,i}(t)}} \lesssim \sqrt{\alpha_{b,i} n_{b,i}(T) \log(T)}.
$$

We have $p(t)$ lesser than $b_i(t)$ because $i$-th buyer participates in round $t$. By UCB property w.h.p. $b_i(t)$ is at most $\sqrt{\frac{\alpha_{b,i} \log(t)}{n_{b,i}(t)}}$ away from $B_i$. A similar argument shows for a non-participating seller $j$ the regret is roughly $\sqrt{\alpha_{s,j} n_{s,j}(T) \log(T)}$.

For a participating buyer $i$, we have the regret bounded as

$$
\sum_{t:i \notin \mathcal{P}_b(t)} (B_i - p^*) + \sum_{t:i \in \mathcal{P}_b(t)} (p(t) - p^*) = (T - n_{b,i}(T))(B_i - p^*) + \sum_{t:i \in \mathcal{P}_b(t)} (p(t) - p^*).
$$

Similarly, a participating seller $j$ has regret bound $(T - n_{s,j}(T))(p^* - S_j) + \sum_{t:j \in \mathcal{P}_s(t)} (p^* - p(t))$.

**Decoupling Participants:** In Lemma 10 in Appendix A, we show that learning is decoupled for the optimal participating buyers, and optimal participating sellers. It lower bounds the number of participation for optimal participating buyers and sellers.

**Lemma 3** (Informal statement of Lemma 10). *For $\alpha_{\min} > 4$, w.h.p., for every $i, j \in [K^*]$, and $i', j' > K^*$*

$$
(T - n_{b,i}(T)) \lesssim \sum_{i'' \geq K^*+1} \frac{\alpha_{b,i''} \log(T)}{(B_i - B_{i''})^2}, \ (T - n_{s,j}(T)) \lesssim \sum_{j'' \geq K^*+1} \frac{\alpha_{s,j''} \log(T)}{(S_{j''} - S_j)^2},
$$

$$
\sum_{t=1}^{T} \mathbb{1}(i \notin \mathcal{P}_b(t), i' \in \mathcal{P}_b(t)) \lesssim \frac{\alpha_{b,i'} \log(T)}{(B_i - B_{i'})^2}, \ and \sum_{t=1}^{T} \mathbb{1}(j \notin \mathcal{P}_s(t), j' \in \mathcal{P}_s(t)) \lesssim \frac{\alpha_{b,j'} \log(T)}{(S_{j'} - S_j)^2}.
$$

It argues after a optimal non-participant $i''$ gets $O(\frac{\log(T)}{(B_i - B_{i''})^2})$ samples it can not participate while $i$ does not participate as the bid of $i$ is higher than the bid of $i''$ with high probability. Similarly seller side result follow.

**Non-Participant regret leads to two-sided Learning:** Unlike optimal participating agents, the effect of uncertainties on the optimal non-participating buyers and sellers cannot be decoupled directly. With at least one non-participating buyer with large estimation error in her valuation present, the non-participating sellers can keep participating even with perfect knowledge. However, this does not ensure directly that the estimation error of this non-participating buyer decreases. Next, in Lemma 12 in Appendix A, we upper bound the number of times a non-participant can participate. Informally we have

**Lemma 4** (Informal). *For $\alpha_{\min} > 4$, with high probability for any $i, j \geq (K^* + 1)$,*

$$n_{b,i}(T) \lesssim \frac{\alpha_{b,i} \log(T)}{(B_{K^*} - B_i)^2} + \sum_{j' \geq (K^*+1)} \frac{\alpha_{s,j'} \log(T)}{(S_{j'} - B_i)^2}, n_{s,j}(T) \lesssim \frac{\alpha_{s,j} \log(T)}{(S_j - S_{K^*})^2} + \sum_{i' \geq (K^*+1)} \frac{\alpha_{b,i'} \log(T)}{(S_j - B_{i'})^2},$$

*and $\sum_{t=1}^{T} \mathbb{1}(j' \in \mathcal{P}_s(t), i' \in \mathcal{P}_b(t)) \lesssim \frac{\alpha_{\max} \log(T)}{(S_{j'} - B_{i'})^2}$.*

A non-participant buyer $i$ after obtaining $O(\frac{\log(T)}{(B_{K^*} - B_i)^2})$ samples does not belong to top $K^*$ w.h.p. Hence, she participates with non-negligible probability only if a seller $j' \geq (K^* + 1)$ participates. However, this implies that with each new match of buyer $i$, additionally at least one non-participating seller is matched, decreasing both their uncertainties. For this buyer $i$, we argue such spurious participation happens a total of $\sum_{j' \geq (K^*+1)} O(\frac{\log(T)}{(S_{j'} - B_i)^2})$ times. After that all the non-participating sellers $j \geq (K^* + 1)$ will have enough samples so that their LCB bids will separate from the $i$-th buyer's UCB bid. Reversing sellers' and buyers' roles does the rest.

**Bounding Price Difference:** The final part of the proof establishes bound on the cumulative difference of price from the true price $p^*$ (see, Lemma 15 in Appendix A).

**Lemma 5.** *(Informal statement of Lemma 15) For $\alpha_{\min} > 4$, w.h.p. $\sum_{t=1}^{T} |p(t) - p^*| \lesssim C \log(T) + \sqrt{\alpha_{max} T \log(T)}$, where $C = O\big(\frac{(M - K^*)(N - K^*)}{\Delta}\big)$.*

Let us focus on the first upper bound, i.e. of the cumulative value of $(p(t) - p^*)$. The proof breaks down the difference into two terms, $2(p(t) - p^*) = (\min_{i \in \mathcal{P}_b(t)} b_i(t) - B_{K^*}) + (S_{K^*} - \max_{j \in \mathcal{P}_s(t)} s_j(t))$. For rounds when the buyer $K^*$ is present (which happens all but $O\big(\frac{(N - K^*) \log(T)}{(B_i - p^*)^2}\big)$ rounds) we can replace $\min_{i \in \mathcal{P}_b(t)} b_i(t)$ with $b_{K^*}(t)$. Finally noticing that $\sum_t (b_{K^*}(t) - B_{K^*}) \lesssim \sqrt{\alpha_{max}} \sqrt{T \log(T)}$ takes care of the first term. For the second term, we need to study the process $\max_{j \in \mathcal{P}_s(t)} s_j(t)$. First we bound the number of times sellers $1$ to $(K^* - 1)$ crosses the seller $K^*$. Next we eliminate all the rounds where at least one seller $j \geq (K^* + 1)$ are participating. Such an elimination comes at a cost of $O\big(\frac{(M - K^*)}{\Delta}\big)$ For any seller $j \geq (K^* + 1)$ this happens $O\big(\frac{(N - K^*) \log(T)}{\Delta_j^2}\big)$ times for some appropriate $\Delta_j$, and gives $\Delta_j$ regret in each round. This final step gives us the dominating $O\big(\frac{(M - K^*)(N - K^*)}{\Delta}\big)$ term. The bound for the cumulative value of $(p^* - p(t))$ follows analogously.

# 5 Lower Bounds

## 5.1 $\Omega(\sqrt{T})$ minimax lower bound on individual regret

We show a minimax regret lower bound of $\Omega(\sqrt{T})$ in Lemma 21 by considering a simpler system that decouples learning and competition. In this system, the seller is assumed to *(i)* know her exact valuation, and *(ii)* always ask her true valuation as the selling price, i.e., is truthful in her asking price in all the rounds. Furthermore, the pricing at every round is fixed to the average $p_t = \frac{B_t + S_t}{2}$ in the event that $B_t \geq S$. We show in Corollary 19 through a coupling argument that any algorithm for the classical two armed bandit problem can be converted to an algorithm for this special case. Then we use well known lower bounds for the bandit problem to give the $\Omega(\sqrt{T})$ lower bound in Lemma 21. All technical details are in Appendix B.

## 5.2 $\Omega(\log(T))$ instance dependent lower bound on Social Welfare Regret

The key observation is that social-welfare regret in Equation (1) is *independent* of the pricing mechanism and only depends on the participating buyers $\mathcal{P}_b(t)$ and sellers $\mathcal{P}_s(t)$ at each time $t$. We will establish a lower bound on a centralized decision maker (DM), who at each time, observes all the rewards obtained by all agents thus far, and decides $\mathcal{P}_b(t)$ and $\mathcal{P}_s(t)$ for each $t$. In Appendix B.7, we show that the actions of the DM can be coupled to that of a combinatorial semi-bandit model [14], where the base set of arms are the set of all buyers and sellers, the valid subset of arms are those subsets having an equal number of buyers and sellers and the mean reward of any valid subset

$\mathcal{A} \subseteq 2^{\mathcal{D}}$ is the difference between the sum of all valuations of buyers in $\mathcal{A}$ and of sellers in $\mathcal{A}$. In Appendix B we exploit this connection to semi-bandits to give a $\Omega(\log(T))$ regret lower bound for the centralized DM. Thus our upper bound of $O(\log(T))$ social welfare regret under the decentralized setting is order optimal since even a centralized system must incur $\Omega(\log(T))$ regret.

## 6 Simulation Study

We perform synthetic studies to augment our theoretical guarantees. For a fixed system of $N$ buyers, $M$ sellers, $K^*$ participants, and $\Delta$ gap, the rewards are Bernoulli, with means themselves chosen uniformly at random. We vary the confidence width of the buyers, $\alpha_b$, and seller, $\alpha_s$, in $[\alpha_1, \alpha_2]$. Next we simulate the performance of the UCB($\alpha_b$) and LCB($\alpha_s$) over 100 independent sample paths with $T = 50k$. We report the mean, 25% and 75% value of the trajectories. We plot the cumulative regret of the buyers, $R_{b,i}(t)$, and the sellers, $R_{s,j}(t)$, the number of matches in the system $K(t)$, and the price difference $(p(t) - p^*)$. In Figure 2, we have a $8 \times 8$ system with $K^* = 5$. We see that $K(t)$ converges to 5, where as $(p(t) - p^*)$ converges to 0. The social regret grows as $log(T)$. The participant and non-participant individual regret of this instance is presented in the appendix in Figure 3. We defer the simulation of other systems used to study behavior of heterogeneous $\alpha$, varying gaps, and different system sizes to Appendix D.

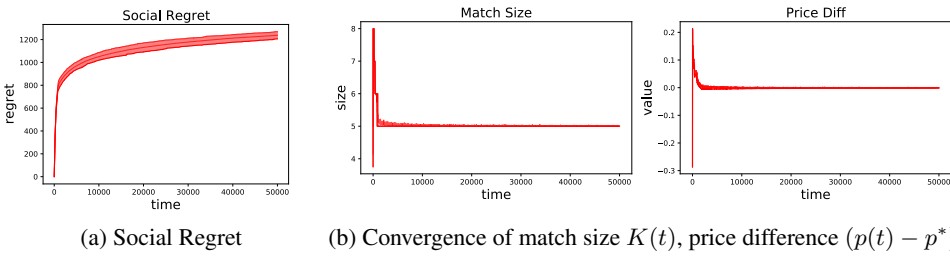

(a) Social Regret   (b) Convergence of match size $K(t)$, price difference $(p(t) - p^*)$

Figure 2: Double Auction $N = 8$, $M = 8$, $K^* = 5$, $\Delta = 0.2$, $\alpha_1 = 4$, and $\alpha_2 = 8$

## 7 Related Work

**Classical mechanism design in double auctions:** There is a large body of work on mechanism design for double auctions, following Myerson et al. [36]. The average-mechanism, which is the subject of focus in this paper achieves all the above desiderata except for being incentive compatible. The VCG mechanism was developed in a series of works [40], [12] and [20] achieves all desiderata except being budget balanced. This mechanism requires the auctioneer to subsidize the trade. More sophisticated trade mechanisms known as the McAfee mechanism [33], trade reduction, and the probabilistic mechanism [5] all trade-off some of the desiderata for others. However, the key assumption in all of these lines of work was that all participants know their own valuations, and do not need to learn through repeated interactions.

**Bandit learning in matching markets:** In recent times, online learning for the two-sided matching markets have been extensively studied in [29], [30], [39], [7], [16]. This line of work studies two sided markets when one of the side does not know of their preferences apriori and learn it through interactions. However, unlike the price-discovery aspect of the present paper, the space of preferences that each participant has to learn is discrete and finite, while the valuations that agents need to learn form a continuum. This model was improved upon by [9] that added notions of price and transfer. The paper of [25] studied a contextual version of the two-sided markets where the agents preferences depend on the reveled context through an unknown function that is learnt through interactions.

**Learning in auctions:** Online learning in simple auctions has a rich history - [6], [13], [34], [8], [15], [41], [22], [17] to name a few, each of which study a separate angle towards learning from repeated samples in auctions. However, unlike the our setting, one side of the market knows of their true valuations apriori. The work of [26] is the closest to ours, where the participants apriori do not know their true valuations. However, they consider the VCG mechanism in the centralized setting, i.e., all participants can observe the utilities of all participants in every round.

**Learning in bi-lateral trade:** One of the first studies on learning in bilateral trade is [10]. The work of [10] considers the single buyer and seller model under a weaker ordinal feedback model, while in the present work we consider the multi-agent model under a stronger bandit feedback model. The feedback model in [10] is more restrictive since the gain from the trade cannot be estimated by the agents based on ordinal feedback while in our model, the gain from the trade can be estimated by each agent. However, our work considers the impact of multi-agent competition on regret minimization, which is not studied in [10]. In each round a buyer and a seller draw her own valuation i.i.d. from their respective distributions. Then an arbiter sets the price, with trade happening if the price is between the seller's and buyer's price. They show with full-information follow-the-leader type algorithm achieves $O(\sqrt{T})$ regret, where as with realistic (bandit-like) feedback by learning the two distributions approximately $O(T^{2/3})$ regret can be achieved.

**Protocol Model:** The 'protocol model' alludes to multiple agents following a similar protocol/algorithm with each agent executing protocol with private information (e.g. Platform-as-a-Service (PAAS) [21]). In the works on learning and markets mentioned here, although the decentralized systems can be modeled as games, the protocol model is studied as a tractable way to capture the essence of the problem [4, 3]. The technical basis for this assumption is that, in the limit when the number of participants are large, and the impact of any single participant is small, a protocol based algorithm is the limit in a precise sense of any equilibrium solution to the multi-agent game [19].

## 8   Conclusion and Future Work

We study the Double auction with Average mechanism where the buyers and sellers need to know their own valuation from her own feedback. Using confidence based bounds – UCB for the buyers and LCB for the sellers, we show that it is possible to obtain $O(\sqrt{T \log(T)})$ individual regret for the true participant buyers and sellers in $T$ rounds. Whereas, the true non-participant buyers and sellers obtain a $O(\log(T)/\Delta)$ individual regret where $\Delta$ is the smallest gap. The social regret of the proposed algorithm also admits a $O(\log(T)/\Delta)$ bound. We show that there are simpler systems where each buyer and seller must obtain a $\Omega(\sqrt{T})$ individual regret in the minimax sense. Moreover, in our setting we show even a centralized controller obtains $\Omega(\log(T))$ social regret. Obtaining a minimax matching a $O(\sqrt{T})$ regret remains open. Another important future avenue is, developing a framework and bidding strategy with provable 'good' regret for general Double auction mechanisms. Alleviating drawbacks of protocol model, such as collusion among participants, and platform disintermediation are important directions to explore.

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
