# A Proofs from Section 4

We setup some notation for the analysis. The total number of samples collected by buyer $i$ is given as $n_{b,i}(t)$, and similarly for seller $j$, it is $n_{s,j}(t)$. Let $K(t)$ be the number of participants in round $t$. For any $i \in [N]$ and $j \in [M]$, let $\chi_{b,i}(t)$ be the indicator of buyer $i$ participating in round $t$, and $\chi_{s,j}(t)$ be the indicator of seller $j$ participating in round $t$. Let $\mathcal{P}_b(t)$ denote the set of participating buyers, and $\mathcal{P}_s(t)$ denote the set of participating sellers.

For any buyer $i \in [N]$ and any time $t$, we have

$$\mathcal{E}_{i,t}^{(\beta)} := \left\{ |\widehat{b}_i(t) - B_i| \leq \sqrt{\frac{\beta \log(t)}{n_{b,i}(t)}} \right\}, \quad \mathbb{P}[\mathcal{E}_{i,t}] \geq 1 - 1/t^{\beta/2}.$$

Similarly, for any seller $j \in [M]$ we have

$$\mathcal{E}_{j,t}^{(\beta)} := \left\{ |\widehat{s}_j(t) - S_j| \leq \sqrt{\frac{\beta \log(t)}{n_{s,j}(t)}} \right\}, \quad \mathbb{P}[\mathcal{E}_{j,t}] \geq 1 - 1/t^{\beta/2}.$$

Without loss of generality, let true valuation of the buyers be in the descending order, and for the seller in the ascending order. Then under the Average mechanism with the true bids (a.k.a. oracle Average mechanism) the buyers $1$ to $K^*$, and the sellers $1$ to $K^*$ participate, while the others do not participate. Let $\alpha_{\min} = \min\{\alpha_{s,j}, \alpha_{b,i} : j \in [M], i \in [N]\}$, and $\alpha_{\max} = \min\{\alpha_{s,j}, \alpha_{b,i} : j \in [M], i \in [N]\}$.

**Lemma 6.** *Under the event $\mathcal{E}_t^{(\beta)} := \cap_{i \in [N]} \cap_{j \in [M]} \mathcal{E}_{i,t}^{(\beta)} \cap \mathcal{E}_{j,t}^{(\beta)}$, in time $t$, the bid for the $i$-th buyer and the $j$-th seller admits the (random) bounds with $\alpha_{\min} \geq \beta$*

$$b_i(t) \in \left[ B_i + (\sqrt{\alpha_{b,i}} - \sqrt{\beta})\sqrt{\frac{\log(t)}{n_{b,i}(t)}}, B_i + (\sqrt{\alpha_{b,i}} + \sqrt{\beta})\sqrt{\frac{\log(t)}{n_{b,i}(t)}} \right],$$

$$s_j(t) \in \left[ S_j - (\sqrt{\alpha_{s,j}} + \sqrt{\beta})\sqrt{\frac{\log(t)}{n_{s,j}(t)}}, S_j - (\sqrt{\alpha_{s,j}} - \sqrt{\beta})\sqrt{\frac{\log(t)}{n_{s,j}(t)}} \right].$$

*In particular, for any buyer $i \in [N]$, any seller $j \in [M]$, and $\alpha_{max} \geq \beta$, we have $b_i(t) \geq B_i$ and $s_j(t) \leq S_j$.*

**Proposition 7.** *For any round $t$, under the event $\mathcal{E}_t^{(\beta)}$, the following events are true, for $\min\{\alpha_{s,j}, \alpha_{b,i}\} \geq \beta$, the number of participants $K(t) \geq K^*$.*

*Proof.* Under the conditions, from Lemma 6 we know that $b_i(t) \geq B_i$ and $s_j(t) \leq S_j$ for all buyers $i$ and sellers $j$. The number of participant in the system given any value profile is given as $K(t) = \max_p \min(|\{i \in [N] : b_i(t) \geq p\}|, |\{j \in [M] : s_j(t) \leq p\}|)$. Let $p^* \in (S_{K^*}, B_{K^*})$. Then we have $|\{i \in [N] : b_i(t) \geq p^*\}| \geq |\{i \in [N] : B_i \geq p^*\}| = K^*$. Similarly, $|\{j \in [M] : s_j(t) \leq p^*\}| \geq |\{j \in [M] : S_j \leq p^*\}| = K^*$. Therefore, we have $K(t) \geq K^*$. $\square$

Let us define the gaps for the seller $j$ from the minimum seller as $\Delta_{s,j} = (S_j - S_{K^*})$, and the gap for the buyer $i$ from the maximum buyer as $\Delta_{b,i} = (B_{K^*} - B_i)$. Also, we define for the seller $j$, $\Delta_{s,b,j} = (S_j - B_{K^*})$, and for the buyer $i$, $\Delta_{b,i} = (S_{K^*} - B_i)$. We have $S_{K^*} \leq B_{K^*}$. We now introduce a definition next to ease exposition of simultaneous participation of two buyers or two sellers.

**Definition 8.** *A buyer $i$ (a seller $j$) precedes a buyer $i'$ (resp., a seller $j'$) if and only if buyer $i$ (resp., seller $j$) participates, and buyer $i'$ (resp., seller $j'$) does not participate.*

The next lemma states that once a true non-participant buyer (seller) have enough samples, this buyer (resp., seller) never precede the true participant buyers (resp., sellers).

**Lemma 9.** *For any round $t$, under the event $\mathcal{E}_t^{(\beta)}$, the following events are true, for $\alpha_{\min} > \beta$*

- *for any specific buyer $i' \in [K^*]$, if for a buyer $i \geq (K^* + 1)$, $n_{b,i}(t) \geq \frac{(\sqrt{\alpha_{b,i}} + \sqrt{\beta})^2}{(B_{i'} - B_i)^2} \log(t)$ then buyer $i$ does not precede buyer $i'$.*

- *for any specific seller $j' \in [K^*]$, if for any $j \geq (K^*+1)$, $n_{s,j}(t) \geq \frac{(\sqrt{\alpha_{s,j}}+\sqrt{\beta})^2}{(S_j - S_{j'})^2} \log(t)$ then seller $j$ does not precede seller $j'$.*

*Proof.* Let $K(t)$ be the number of participants in round $t$. We know that under $\mathcal{E}_t^{(\beta)}$ and $\alpha_{\min} > \beta$, $s_j(t) < S_j$ and $b_i(t) > B_i$ for all $i \in [N]$ and $j \in [M]$. We note that if for any $i \geq (K^*+1)$ and $i' \leq K^*$, if $n_{b,i}(t) \geq \frac{(\sqrt{\alpha_{b,i}}+\sqrt{\beta})^2}{(B_{i'} - B_i)^2} \log(t)$ then $b_i(t) \leq B_{i'}$. For buyers $i'$ we have $b_{i'}(t) > B_{i'}$, hence buyer $i$ can not precede any of the buyers $i'$. This is true as under the current mechanism for any $i'$ with $b_{i'}(t) > b_i(t)$, it can not happen that buyer $i'$ participates but buyer $i$ does not. A similar argument proves the seller side statement. $\square$

Furthermore, the above Lemma 9 can be used in conjunction to Proposition 7, to show the true participant buyers (or sellers) fail to match only logarithimically many times.

**Lemma 10.** *Under the event $\mathcal{E}^{(\beta)} = \cup_{t=1}^T \mathcal{E}_t^{(\beta)}$ and $\alpha_{\min} > \beta$, we have*

- $n_{b,i}(T) \geq T - \sum_{i' \geq K^*+1} \frac{(\sqrt{\alpha_{b,i'}}+\sqrt{\beta})^2}{(B_i - B_{i'})^2} \log(T)$ *for any $i \in [K^*]$,*

- $n_{s,j}(T) \geq T - \sum_{j' \geq K^*+1} \frac{(\sqrt{\alpha_{s,j'}}+\sqrt{\beta})^2}{(S_{j'} - S_j)^2} \log(T)$ *for any $j \in [K^*]$.*

*Proof.* We know that under the condition of the lemma, $K(t) \geq K^*$ for all $1 \leq t \leq T$. Therefore, we have $\sum_{i \in [N]} n_{b,i}(T) \geq K^* T$. Furthermore, as $K(t) \geq K^*$ in each round, a buyer $i \in [K^*]$ does not participate, only if there exists at least one participant $i' \geq (K^*+1)$ that precedes buyer $i$. This is because if in some round no participant $i' \geq (K^*+1)$ precedes buyer $i$ and buyer $i$ does not match then that implies at most there can be $(K^*-1)$ matches in that round. This leads to a contradiction of $K(t) \geq K^*$. However, under event $\mathcal{E}^{(\beta)}$ and $\alpha_{\min} > \beta$, from Lemma 9 we know that any $i' \geq (K^*+1)$ can precede buyer $i$ only $\frac{(\sqrt{\alpha_{b,i'}}+\sqrt{\beta})^2}{(B_i - B_{i'})^2} \log(T)$ many times. This implies that $(T - n_{b,i}(T)) \leq \sum_{i' \geq K^*+1} \frac{(\sqrt{\alpha_{b,i'}}+\sqrt{\beta})^2}{(B_i - B_{i'})^2} \log(T)$.

A similar treatment of the sellers give us the remaining result. $\square$

We next show that the true non-participants only match logarithimically many times. This means $K(t)$ converges to $K^*$ fast. It is important to note that the previous argument does not conclude that where we mainly relied upon $K(t) \geq K^*$ for all $t$ w.h.p. To that end we introduce the following definition.

**Definition 11.** *A buyer $i$, and a seller $j$ co-participates in a given round, if and only if both buyer $i$, and seller $j$ participates in that round.*

**Lemma 12.** *Under the event $\mathcal{E}^{(\beta)} = \cup_{t=1}^T \mathcal{E}_t^{(\beta)}$ and $\alpha_{\min} > \beta$, we have*

- $\sum_{t=1}^T \mathbb{1}(j' \in \mathcal{P}_s(t), i' \in \mathcal{P}_b(t)) \leq \frac{(\sqrt{\alpha_{\max}}+\sqrt{\beta})^2}{(S_{j'} - B_{i'})^2} \log(T)$ *for any $i', j' \geq (K^*+1)$*

- $n_{b,i}(T) \leq \left( \frac{(\sqrt{\alpha_{b,i}}+\sqrt{\beta})^2}{(B_{K^*} - B_i)^2} + \sum_{j \geq (K^*+1)} \frac{(\sqrt{\alpha_{s,j}}+\sqrt{\beta})^2}{(S_j - B_i)^2} \right) \log(T)$ *for any $i \geq (K^*+1)$ and* $\sum_{i \geq (K^*+1)} n_{b,i}(T) \leq \left( \sum_{i \geq (K^*+1)} \frac{(\sqrt{\alpha_{b,i}}+\sqrt{\beta})^2}{(B_{K^*} - B_i)^2} + \sum_{j \geq (K^*+1)} \frac{(\sqrt{\alpha_{s,j}}+\sqrt{\beta})^2}{(S_j - B_{K^*})^2} \right) \log(T)$,

- $n_{s,j}(T) \leq \left( \frac{(\sqrt{\alpha_{s,j}}+\sqrt{\beta})^2}{(S_j - S_{K^*})^2} + \sum_{i \geq (K^*+1)} \frac{(\sqrt{\alpha_{b,i}}+\sqrt{\beta})^2}{(S_j - B_i)^2} \right) \log(T)$ *for any $j \geq (K^*+1)$, and* $\sum_{j \geq (K^*+1)} n_{s,j}(T) \leq \left( \sum_{j \geq (K^*+1)} \frac{(\sqrt{\alpha_{s,j}}+\sqrt{\beta})^2}{(S_j - S_{K^*})^2} + \sum_{i \geq (K^*+1)} \frac{(\sqrt{\alpha_{b,i}}+\sqrt{\beta})^2}{(S_{K^*} - B_i)^2} \right) \log(T)$.

*Proof.* Let us consider a non-participant buyer $i$. From Lemma 9 we know that if $n_{b,i}(t) \geq Th_i(t) \equiv \frac{(\sqrt{\alpha_{b,i}}+\sqrt{\beta})^2}{(B_{K^*} - B_i)^2} \log(t)$ then buyer $i$ does not precede any buyer $i' \in [K^*]$. Therefore, for this buyer to match there should exist at least $(K^*+1)$ sellers with bids no more than the bid of this buyer $i$, and at least $(K^*+1)$ seller participates. This further implies that for this buyer to have additional participation, after $n_{b,i}(t) = Th_i(T)$:

1. There exists at least 1 seller $j \geq (K^* + 1)$ with bid no more than the bid of this buyer $i$.

2. There exists at least 1 seller $j \geq (K^* + 1)$ such that buyer $i$ and seller $j$ co-participates.

Let $\mathcal{S}_{i,j}$ be the rounds after $n_{b,i}(t) = Th_i(T)$, and buyer $i$ and seller $j$ co-participates. Therefore, from the 2nd point above we conclude that $n_{b,i}(T) \leq Th_i(T) + |\cup_{j \geq (K^*+1)} \mathcal{S}_{i,j}|$.

However, if the seller $j$ has $n_{s,j}(t) > Th_{i,j}(t) \equiv \frac{(\sqrt{\alpha_{s,j}}+\sqrt{\beta})^2}{(S_j - B_i)^2} \log(t)$ many participation then under $\mathcal{E}_t^{(\beta)}$ and $\alpha_{\min} > \beta$ we have $s_j(t) > b_i(t)$.[4] But then from the 1st point above we know that $|\mathcal{S}_{i,j}| \leq Th_{i,j}(T)$, because $|\mathcal{S}_{i,j}| \leq n_{s,j}(T)$, and buyer $i$ and seller $j$ can co-participate only if $n_{s,j}(t) \leq Th_{i,j}(T)$. This proves the first point. Moreover, we have $n_{b,i}(T) \leq Th_i(T) + \sum_{j \geq (K^*+1)} Th_{i,j}(T)$.

In fact, we can improve the cumulative bound. We have

$$\sum_{i \geq (K^*+1)} n_{b,i}(T) \leq \sum_{i \geq (K^*+1)} Th_i(T) + |\cup_{i \geq (K^*+1)} \cup_{j \geq (K^*+1)} \mathcal{S}_{i,j}|.$$

However, we know that after $n_{s,j}(t) > \max_i Th_{i,j}(T)$ a seller $j \geq (K^* + 1)$ can not co-participate for any seller $i \geq (K^* + 1)$. Thus we can bound $|\cup_{i \geq (K^*+1)} \mathcal{S}_{i,j}| \leq \max_i Th_{i,j}(T)$.

$$\sum_{i \geq (K^*+1)} n_{b,i}(T) \leq \sum_{i \geq (K^*+1)} Th_i(T) + \sum_{j \geq (K^*+1)} \max_{i \geq (K^*+1)} Th_{i,j}(T).$$

A similar treatment proves the lemma for a non-participant seller $j$. $\square$

**Corollary 13.** *Under the event $\mathcal{E}^{(\beta)} = \cup_{t=1}^T \mathcal{E}_t^{(\beta)}$ and $\alpha_{\min} > \beta$, we have*

- $n_{b,i}(T) \leq \frac{(M-K^*+1)(\sqrt{\alpha_{\max}}+\sqrt{\beta})^2}{(B_{K^*}-B_i)^2} \log(T)$ *for any $i \geq (K^* + 1)$,*

- $n_{s,j}(T) \leq \frac{(N-K^*+1)(\sqrt{\alpha_{\max}}+\sqrt{\beta})^2}{(S_j-S_{K^*})^2} \log(T)$ *for any $j \geq (K^* + 1)$.*

We have shown, up to this point, that the optimal participating buyers and sellers participate in all but $O(log(T))$ rounds. Moreover, true non-participating buyers and sellers participate in $O(log(T))$ rounds. This suffices to show the regret for non-participating buyers and sellers is $O(log(T))$ (we will state this precisely later). However, to compute the regret for participating buyers and sellers we next need to understand how the price is set in each round. We next argue that if in any round $t$, sellers $j \geq (K^* + 1)$ do not participate, and buyer $K^*$ participates then the regret of a optimal participating buyer is small. Similarly, if in any round $t$, buyers $i \geq (K^* + 1)$ do not participate, and seller $K^*$ participates then the regret of a optimal participating seller is small.

### A.1  Social Welfare regret of buyers and sellers.

We now compute the social welfare regret.

**Lemma 14.** *Under the event $\mathcal{E}^{(\beta)} = \cup_{t=1}^T \mathcal{E}_t^{(\beta)}$ and $\alpha_{\min} > \beta$, we have the social regret*

$$r_{SW}(T) \leq \sum_{i \leq K^*} \sum_{i' > K^*} \frac{(\sqrt{\alpha_{\max}}+\sqrt{\beta})^2}{(B_i - B_{i'})} \log(T) + \sum_{j \leq K^*} \sum_{j' > K^*} \frac{(\sqrt{\alpha_{\max}}+\sqrt{\beta})^2}{(S_{j'} - S_j)} \log(T)$$

$$+ \sum_{j' > K^*} \sum_{i' > K^*} \frac{(\sqrt{\alpha_{\max}}+\sqrt{\beta})^2}{(S_{j'} - B_{i'})}.$$

*Proof.* Let us consider that the event $\mathcal{E}^{(\beta)} = \cup_{t=1}^T \mathcal{E}_t^{(\beta)}$ holds, and $\min\{\alpha_{s,j}, \alpha_{b,i}\} \geq \beta$. With that assumption we can bound the social welfare regret as follows.

---

[4]Note that for any $i, j \geq (K^* + 1)$ we have $S_j > B_i$.

$$r_{SW}(T) = T\Big(\sum_{i\in\mathcal{P}_b^*} B_i + \sum_{j\in[M]\setminus\mathcal{P}_s^*} S_j\Big) - \Big[\sum_{t=1}^{T}\Big(\sum_{i\in\mathcal{P}_b(t)} B_i + \sum_{j\in[M]\setminus\mathcal{P}_s(t)} S_j\Big)\Big]$$

$$= \sum_{t=1}^{T}\Big[\Big(\sum_{i\in[K^*]\setminus\mathcal{P}_b(t)} B_i - \sum_{i'\in\mathcal{P}_b(t)\setminus[K^*]} B_{i'}\Big) + \Big(\sum_{j'\in\mathcal{P}_s(t)\setminus[K^*]} S_{j'} - \sum_{j\in[K^*]\setminus\mathcal{P}_s(t)} S_j\Big)\Big]$$

We use $\mathcal{P}_b^* = \mathcal{P}_b^* = [K^*]$ without loss of generality, as mentioned earlier.

We first notice that above the number of positive and negative terms are equal. Now we consider pairing some of the positive and negative terms. Under the event $\mathcal{E}^{(\beta)} = \cup_{t=1}^{T}\mathcal{E}_t^{(\beta)}$ and $\min\{\alpha_{s,j}, \alpha_{b,i}\} \geq \beta$, we know that $K(t) \geq K^*$ where $\mathcal{P}_b(t) = \mathcal{P}_s(t) = K(t)$. Hence, we have

$$|[K^*]\setminus\mathcal{P}_b(t)| \leq |\mathcal{P}_b(t)\setminus[K^*]|, \qquad |[K^*]\setminus\mathcal{P}_s(t)| \leq |\mathcal{P}_s(t)\setminus[K^*]|.$$

Therefore, in the final term we can *pair up* each buyer with positive contribution with a buyer with negative contribution, and each seller with negative contribution with a seller with positive contribution. Finally, we may be left with some sellers with positive, and some buyers with negative contributions. But we have the number of sellers with positive contribution equals the number of buyers with negative contribution. To see this observe

$$(|\mathcal{P}_b(t)\setminus[K^*]| - |[K^*]\setminus\mathcal{P}_b(t)|) = (|\mathcal{P}_s(t)\setminus[K^*]| - |[K^*]\setminus\mathcal{P}_s(t)|) = (K(t) - K^*)$$

Let us define by $i'(i,t) \in \mathcal{P}_b(t)\setminus[K^*]$ as the pair for the buyer $i \in [K^*]\setminus\mathcal{P}_b(t)$, such that $i'(i,t)$ are all unique. Similarly, we denote by $j'(j,t) \in \mathcal{P}_s(t)\setminus[K^*]$ the pair for $j \in [K^*]\setminus\mathcal{P}_s(t)$. We denote the set of remaining buyers and sellers, respectively, as

$$\mathcal{I}'(t) = (\mathcal{P}_b(t)\setminus[K^*])\setminus\cup_{i\in[K^*]\setminus\mathcal{P}_b(t)}i'(i,t),$$
$$\mathcal{J}'(t) = (\mathcal{P}_s(t)\setminus[K^*])\setminus\cup_{j\in[K^*]\setminus\mathcal{P}_s(t)}j'(j,t).$$

Finally, we denote by $i(j',t) \in \mathcal{I}'(t)$ as the pair for $j' \in \mathcal{J}'(t)$.[5]

With these definitions we can bound the social regret as

$$r_{SW}(T) = \sum_{t=1}^{T}\Big[\sum_{i\in[K^*]\setminus\mathcal{P}_b(t)}(B_i - B_{i'(i,t)}) + \sum_{j\in[K^*]\setminus\mathcal{P}_s(t)}(S_{j'(j,t)} - S_j) + \sum_{j\in\mathcal{J}'(t)}(S_j - B_{i(j',t)})\Big]$$

$$= \sum_{t=1}^{T}\Big[\sum_{i\leq K^*}(B_i - B_{i'(i,t)})\mathbb{1}(i'(i,t)\in\mathcal{P}_b(t), i\notin\mathcal{P}_b(t))$$

$$+ \sum_{j\leq K^*}(S_{j'(j,t)} - S_j)\mathbb{1}(j'(j,t)\in\mathcal{P}_s(t), j\notin\mathcal{P}_s(t))$$

$$+ \sum_{j'>K^*}(S_{j'} - B_{i(j',t)})\mathbb{1}(j'\in\mathcal{P}_s(t), i(j',t)\in\mathcal{P}_b(t))\Big]$$

$$\leq \sum_{t=1}^{T}\Big[\sum_{i\leq K^*}\sum_{i'>K^*}(B_i - B_{i'})\mathbb{1}(i'\in\mathcal{P}_b(t), i\notin\mathcal{P}_b(t))$$

$$+ \sum_{j\leq K^*}\sum_{j'>K^*}(S_{j'} - S_j)\mathbb{1}(j'\in\mathcal{P}_s(t), j\notin\mathcal{P}_s(t))$$

$$+ \sum_{i'>K^*}\sum_{j'>K^*}(S_{j'} - B_{i'})\mathbb{1}(j'\in\mathcal{P}_s(t), i'\in\mathcal{P}_b(t))\Big] \tag{3}$$

$$= \sum_{i\leq K^*}\sum_{i'>K^*}(B_i - B_{i'})\sum_{t=1}^{T}\mathbb{1}(i'\in\mathcal{P}_b(t), i\notin\mathcal{P}_b(t))$$

---

[5]Any arbitrary pairing works for this purpose. For concreteness, we may assume the the buyer/seller in one set, is matched with that of the other ranked by their respective ids.

$$+ \sum_{j \leq K^*} \sum_{j' > K^*} (S_{j'} - S_j) \sum_{t=1}^{T} \mathbb{1}(j' \in \mathcal{P}_s(t), j \notin \mathcal{P}_s(t))$$

$$+ \sum_{i' > K^*} \sum_{j' > K^*} (S_{j'} - B_{i'}) \sum_{t=1}^{T} \mathbb{1}(j' \in \mathcal{P}_s(t), i' \in \mathcal{P}_b(t))$$

$$\leq \sum_{i \leq K^*} \sum_{i' > K^*} \frac{(\sqrt{\alpha_{b,i'}} + \sqrt{\beta})^2}{(B_i - B_{i'})} \log(T) + \sum_{j \leq K^*} \sum_{j' > K^*} \frac{(\sqrt{\alpha_{s,j'}} + \sqrt{\beta})^2}{(S_{j'} - S_j)} \log(T)$$

$$+ \sum_{j' > K^*} \sum_{i' > K^*} \frac{(\sqrt{\alpha_{s,i'}} + \sqrt{\beta})^2}{(S_{j'} - B_{i'})} \log(T). \tag{4}$$

The first inequality (3) upper bounds $i'(i,t)$, $j'(j,t)$, and $i(j',t)$ with the sum over the sets each of them can belong to.

Under the event $\mathcal{E}^{(\beta)} = \cup_{t=1}^T \mathcal{E}_t^{(\beta)}$ and $\min\{\alpha_{s,j}, \alpha_{b,i}\} \geq \beta$, the final inequality (4) follows from Lemma 9 and Lemma 12. In particular, $\sum_{t=1}^T \mathbb{1}(i' \in \mathcal{P}_b(t), i \notin \mathcal{P}_b(t))$ denotes the number of times a true non-participating buyer $i'$ can precede a optimal participating buyer $i$, and Lemma 9 bounds these terms. Similarly, the terms $\sum_{t=1}^T \mathbb{1}(j' \in \mathcal{P}_s(t), j \notin \mathcal{P}_s(t))$ are bounded with the help of Lemma 9. Finally, the terms $\mathbb{1}(j' \in \mathcal{P}_s(t), i' \in \mathcal{P}_b(t))$ denote how many times a pair of true non-participating buyer and seller $i'$ and $j'$ can co-participate. Following proof of Lemma 12, we can bound this with $\frac{(\sqrt{\alpha_{s,i'}} + \sqrt{\beta})^2}{(S_{j'} - B_{i'})} \log(T)$. This finishes the proof of the lemma. $\square$

Using Lemma 14, the expected regret can be bounded for $\beta > 2$ as

$$R_{SW}(T) \leq \mathbb{E}[r_{SW}(T) | \mathcal{E}^{(\beta)}] + b_{max}(1 - \mathbb{P}[\mathcal{E}^{(\beta)}])$$

$$\leq \mathbb{E}[r_{SW}(T) | \mathcal{E}^{(\beta)}] + b_{max} \sum_{t=1}^{T} MN/t^{\beta/2}$$

$$\leq \mathbb{E}[r_{SW}(T) | \mathcal{E}^{(\beta)}] + MN b_{max} \zeta(\beta/2).$$

Here $\zeta(x)$ is the Riemann-zeta function which is finite for $x > 1$.

### A.2 Individual regret of buyers and sellers.

Recall that $p^* = (S_{K^*} + B_{K^*})/2$ be the price under true bids for the average mechanism, and $p(t) = (\min_{i' \in \mathcal{P}_b(t)} b_{i'}(t) + \max_{j \in \mathcal{P}_s(t)} s_j(t))/2$ denotes the price in round $t$. Let $\chi_{b,i}(t)$, and $\chi_{s,j}(t)$ is the participation indicator for buyer $i$, and seller $j$ respectively. The individual regret for any true non-participating buyer or seller can be computed easily. We will present it later.

The regret for any optimal participating buyer $i \in [K^*]$ can be decomposed in rounds where the buyer $i$ does not participate, and where the buyer $i$ participates.

$$r_{b,i}(T) = \sum_{t:\chi_{b,i}(t)=0} (B_i - (B_{K^*} + S_{K^*})/2) + \sum_{t:\chi_{b,i}(t)=1} (p(t) - p^*).$$

Similarly, for any optimal participating seller $j \in [K^*]$ the regret is bounded as

$$r_{s,j}(T) = \sum_{t:\chi_{s,j}(t)=0} ((B_{K^*} + S_{K^*})/2 - S_j) + \sum_{t:\chi_{s,j}(t)=1} (p^* - p(t)).$$

We first focus on the regret of the buyers which implies upper bounding $\sum_t (p(t) - p^*)$ in the next lemma.

**Lemma 15.** *Under the event* $\mathcal{E}^{(\beta)} = \cup_{t=1}^T \mathcal{E}_t^{(\beta)}$ *and* $\alpha_{\min} > \beta$, *we have for all* $\tilde{i} \in [N]$ *and* $\tilde{j} \in [M]$

$$\sum_{t:\chi_{b,\tilde{i}}(t)=1} (p(t) - p^*) \leq C_b \log(T) + (\sqrt{\alpha_{max}} + \sqrt{\beta}) \sqrt{n_{b,\tilde{i}}(T) \log(T)},$$

$$\sum_{t:\chi_{s,\tilde{j}}(t)=1} (p^* - p(t)) \leq C_s \log(T) + (\sqrt{\alpha_{max}} + \sqrt{\beta})\sqrt{n_{b,\tilde{j}}(T)\log(T)},$$

*where*

$$2C_b = \sum_{j<K^*} \frac{(\sqrt{\alpha_{max}} + \sqrt{\beta})^2}{(S_{K^*} - S_j)} + \sum_{i\geq(K^*+1)} \frac{(\sqrt{\alpha_{max}} + \sqrt{\beta})^2\sqrt{(M-K^*+1)}}{(B_{K^*} - B_i)}$$

$$+ \sum_{j\geq(K^*+1)} \left( \frac{(N-K^*+1)(\sqrt{\alpha_{max}} + \sqrt{\beta})^2}{(S_j - S_{K^*})} + \frac{\sqrt{(N-K^*+1)}(\sqrt{\alpha_{max}} + \sqrt{\beta})^2}{(S_j - S_{K^*})} \right),$$

$$2C_s = \sum_{i<K^*} \frac{(\sqrt{\alpha_{max}} + \sqrt{\beta})^2}{(B_i - B_{K^*})} + \sum_{j\geq(K^*+1)} \frac{(\sqrt{\alpha_{max}} + \sqrt{\beta})^2\sqrt{(N-K^*+1)}}{(S_j - S_{K^*})}$$

$$+ \sum_{i\geq(K^*+1)} \left( \frac{(M-K^*+1)(\sqrt{\alpha_{max}} + \sqrt{\beta})^2}{(B_{K^*} - B_i)} + \frac{\sqrt{(M-K^*+1)}(\sqrt{\alpha_{max}} + \sqrt{\beta})^2}{(B_{K^*} - B_i)} \right).$$

We are now in a position to prove out main theorem.

**Theorem 16.** *The regret of the Average mechanism with buyers bidding UCB($\alpha$), and sellers bidding LCB($\alpha$) of their estimated valuation, for $\alpha_{\min} > \beta > 2$, we have the expected regret is bounded as:*

- *for a participating buyer $i \in [K^*]$ as $R_{b,i}(T) \leq (\sqrt{\alpha_{max}} + \sqrt{\beta})\sqrt{T\log(T)} + C_{b',i}\log(T)$,*

- *for a participating seller $j \in [K^*]$ as $R_{s,j}(T) \leq (\sqrt{\alpha_{max}} + \sqrt{\beta})\sqrt{T\log(T)} + C_{s',j}\log(T)$,*

- *for a non-participating buyer $i \geq (K^*+1)$ as*
  $R_{b,i}(T) \leq \frac{\sqrt{(M-K^*+1)}(\sqrt{\alpha_{max}} + \sqrt{\beta})^2}{(B_{K^*} - B_i)} \log(T)$,

- *for a non-participating seller $j \geq (K^*+1)$ as*
  $R_{s,j}(T) \leq \frac{\sqrt{(N-K^*+1)}(\sqrt{\alpha_{max}} + \sqrt{\beta})^2}{(S_j - S_{K^*})} \log(T)$.

*Here $C_b$ and $C_s$ is as defined in Lemma 15, and*

$$C_{b',i} = \left( (N-K^*)\frac{(\sqrt{\alpha_{max}} + \sqrt{\beta})^2}{(B_i - p^*)} + C_b \right),$$

$$C_{s',j} = \left( (M-K^*)\frac{(\sqrt{\alpha_{max}} + \sqrt{\beta})^2}{(p^* - S_j)} + C_s \right).$$

*Proof of Regret Upper Bound.* We first bound the regret under the event $\mathcal{E}^{(\beta)} = \cup_{t=1}^{T}\mathcal{E}_t^{(\beta)}$ and $\alpha_{\min} > \beta$. Applying the bounds in Lemma 15, we bound the of a optimal participating buyer $i \in [K^*]$ as

$$r_{b,i}(T) = \sum_{t:\chi_{b,i}(t)=0} (B_i - (B_{K^*} + S_{K^*})/2) + \sum_{t:\chi_{b,i}(t)=1} (p(t) - p^*)$$

$$\leq \sum_{i'\geq K^*+1} \frac{(\sqrt{\alpha_{max}} + \sqrt{\beta})^2(B_i - (B_{K^*} + S_{K^*})/2)}{(B_i - B_{i'})^2} \log(T) + C_b \log(T) + (\sqrt{\alpha_{max}} + \sqrt{\beta})\sqrt{T\log(T)}$$

$$\leq \left( (N-K^*)\frac{(\sqrt{\alpha_{max}} + \sqrt{\beta})^2}{(B_i - p^*)} + C_b \right)\log(T) + (\sqrt{\alpha_{max}} + \sqrt{\beta})\sqrt{T\log(T)}$$

Similarly, for any optimal participating seller $j \in [K^*]$ the regret is bounded as

$$r_{s,j}(T) = \sum_{t:\chi_{s,j}(t)=0} ((B_{K^*} + S_{K^*})/2 - S_j) + \sum_{t:\chi_{s,j}(t)=1} (p^* - p(t))$$

$$\leq \sum_{j'\geq K^*+1} \frac{(\sqrt{\alpha_{max}} + \sqrt{\beta})^2((B_{K^*} + S_{K^*})/2 - S_j)}{(S_{j'} - S_j)^2} \log(T) + C_s \log(T) + (\sqrt{\alpha_{max}} + \sqrt{\beta})\sqrt{T\log(T)}$$

$$\leq \left( (M - K^*) \frac{(\sqrt{\alpha_{max}} + \sqrt{\beta})^2}{(p^* - S_j)} + C_s \right) \log(T) + (\sqrt{\alpha_{max}} + \sqrt{\beta})\sqrt{T \log(T)}$$

The regret for any true non-participating buyer $i \geq (K^* + 1)$ is non negative only when the buyer $i$ participates. Under $\mathcal{E}^{(\beta)}$ and $\alpha_{\min} > \beta$ we have

$$r_{b,i}(T) = \sum_{t:\chi_{b,i}(t)=1} (p(t) - B_i)$$

$$\leq \sum_{t:\chi_{b,i}(t)=1} (b_i(t) - B_i)$$

$$\leq \sum_{t:\chi_{b,i}(t)=1} (\sqrt{\alpha_{max}} + \sqrt{\beta})\sqrt{\frac{log(t)}{n_{b,i}(t)}}$$

$$\leq \sum_{n=1}^{n_{b,i}(T)} (\sqrt{\alpha_{max}} + \sqrt{\beta})\sqrt{\frac{log(T)}{n}}$$

$$\leq (\sqrt{\alpha_{max}} + \sqrt{\beta})\sqrt{n_{b,i}(T) \log(T)}$$

Where the first inequality is due to the fact that if buyer $i$ participates in round $t$ then bid $b_i(t) \geq p(t)$.

Also, the regret for any true non-participating buyer $j \geq (K^* + 1)$ is non negative only when the buyer $i$ participates. Under $\mathcal{E}^{(\beta)}$ and $\alpha_{\min} > \beta$ we have similarly

$$r_{s,j}(T) = \sum_{t:\chi_{s,j}(t)=1} (S_j - s_j(t)) \leq (\sqrt{\alpha_{max}} + \sqrt{\beta})\sqrt{n_{s,j}(T) \log(T)}$$

The terms $n_{b,i}(T)$ and $n_{s,j}(T)$ above can be bounded using Lemma 12 when $\mathcal{E}^\beta$ holds for $\alpha_{max} \geq \beta$

Therefore, the expected regret can be bounded for $\beta > 2$ as

$$R_{b,i}(T) \leq \mathbb{E}[r_{b,i}(T)|\mathcal{E}^{(\beta)}] + b_{max}(1 - \mathbb{P}[\mathcal{E}^{(\beta)}])$$

$$\leq \mathbb{E}[r_{b,i}(T)|\mathcal{E}^{(\beta)}] + b_{max} \sum_{t=1}^{T} MN/t^{\beta/2}$$

$$\leq \mathbb{E}[r_{b,i}(T)|\mathcal{E}^{(\beta)}] + MNb_{max}\zeta(\beta/2).$$

Similarly, for $\beta > 2$ we have

$$R_{s,j}(T) \leq \mathbb{E}[r_{s,j}(T)|\mathcal{E}^{(\beta)}] + MNs_{max}\zeta(\beta/2).$$

This concludes the proof. $\qquad\square$

Let us recall the minimum gap is $\Delta = \min_{i,\in[N],j\in[M]}\{|p^* - S_j|, |B_i - p^*|\}$. Then we can bound the constants associated with the logarithmic terms as

**Corollary 17.** *The constants in Theorem 16 is upper bounded as*

- $C_{b'} \leq \frac{\left(N+(N-K^*+1)\sqrt{M-K^*+1}+\sqrt{N-K^*+1}(M-K^*+1)+(N-K^*+1)(M-K^*+1)\right)(\sqrt{\alpha_{max}}+\sqrt{\beta})^2}{\Delta}$

- $C_{s'} \leq \frac{\left(M+(N-K^*+1)\sqrt{M-K^*+1}+\sqrt{N-K^*+1}(M-K^*+1)+(N-K^*+1)(M-K^*+1)\right)(\sqrt{\alpha_{max}}+\sqrt{\beta})^2}{\Delta}$

### A.3 Proof of Lemma 15

*Proof.* We first focus on the upper bound for some buyer $i \in [N]$.

$$\sum_{t:\chi_{b,\bar{i}}(t)=1} (p(t) - p^*) = \frac{1}{2} \sum_{t:\chi_{b,\bar{i}}(t)=1} \left( \min_{i' \in \mathcal{P}_b(t)} b_{i'}(t) - B_{K^*} \right) + \frac{1}{2} \sum_{t:\chi_{b,\bar{i}}(t)=1} \left( \max_{j \in \mathcal{P}_s(t)} s_j(t) - S_{K^*} \right)$$

The rounds where buyer $K^*$ participates, we have $\min_{i \in \mathcal{P}_b(t)} b_i(t) \leq b_{K^*}(t)$. Therefore, we can bound

$$\sum_t \left( \min_{i' \in \mathcal{P}_b(t)} b_{i'}(t) - B_{K^*} \right)$$

$$\leq \sum_{t: \chi_{b,K^*}(t)=0} \left( \min_{i' \in \mathcal{P}_b(t)} b_{i'}(t) - B_{K^*} \right) + \sum_{t: \chi_{b,K^*}(t)=1, \chi_{b,\tilde{i}}(t)=1} (b_{K^*}(t) - B_{K^*})$$

Under $\mathcal{E}^{(\beta)} = \cup_{t=1}^T \mathcal{E}_t^{(\beta)}$ and $\alpha_{\min} > \beta$, we know that $\chi_{b,K^*}(t) = 0$ only if $\max_{i \geq (K^*+1)} \chi_{b,K^*}(t) = 1$. That is $K^*$ does not participate, only if at least one of the non-participant buyers participate. Hence we further have,

$$\sum_{t: \chi_{b,K^*}(t)=0} \left( \min_{i' \in \mathcal{P}_b(t)} b_{i'}(t) - B_{K^*} \right)$$

$$\leq \sum_{i \geq (K^*+1)} \sum_{t: \chi_{b,i}(t)=1, \chi_{b,K^*}(t)=0} (b_i(t) - B_{K^*})$$

$$\leq \sum_{i \geq (K^*+1)} \sum_{t: \chi_{b,i}(t)=1} (b_i(t) - B_{K^*})$$

$$\leq \sum_{i \geq (K^*+1)} \sum_{t: \chi_{b,i}(t)=1} (b_i(t) - B_i)$$

Under the event $\mathcal{E}^{(\beta)} = \cup_{t=1}^T \mathcal{E}_t^{(\beta)}$ and $\alpha_{\min} > \beta$, we have

$$\sum_{t: \chi_{b,K^*}(t)=1, \chi_{b,\tilde{i}}(t)=1} (b_{K^*}(t) - B_{K^*})$$

$$\leq \sum_{t: \chi_{b,K^*}(t)=1, \chi_{b,\tilde{i}}(t)=1} (\sqrt{\alpha_{b,K^*}} + \sqrt{\beta}) \sqrt{\frac{\log(t)}{n_{b,K^*}(t)}}$$

$$\leq \sum_{n=1}^{n_{b,\tilde{i}}(T)} (\sqrt{\alpha_{b,K^*}} + \sqrt{\beta}) \sqrt{\frac{\log(T)}{n}} \leq (\sqrt{\alpha_{b,K^*}} + \sqrt{\beta}) \sqrt{n_{b,\tilde{i}}(T) \log(T)},$$

Above, we use the logic that the summation is minimized when $n_{b,K^*}(t)$ increases in unison with $n_{b,i}(t)$, otherwise we will have larger denominator.

From Corollary 13 we know that under the event $\mathcal{E}^{(\beta)}$ and $\alpha_{\min} > \beta$, the maximum number of time a buyer $i \geq (K^*+1)$ can participate is

$$\tilde{T}h_i(T) = (M - K^* + 1) \frac{(\sqrt{\alpha_{\max}} + \sqrt{\beta})^2}{(B_{K^*} - B_i)^2} \log(T).$$

Under the event $\mathcal{E}^{(\beta)} = \cup_{t=1}^T \mathcal{E}_t^{(\beta)}$ and $\alpha_{\min} > \beta$, we have

$$\sum_{t: \chi_{b,i}(t)=1} (b_i(t) - B_i)$$

$$\leq \sum_{t: \chi_{b,i}(t)=1} (\sqrt{\alpha_{\max}} + \sqrt{\beta}) \sqrt{\frac{\log(t)}{n_{b,i}(t)}}$$

$$\leq \sum_{n=1}^{\tilde{T}h_i(T)} (\sqrt{\alpha_{\max}} + \sqrt{\beta}) \sqrt{\frac{\log(T)}{n}}$$

$$\leq (\sqrt{\alpha_{\max}} + \sqrt{\beta}) \sqrt{\tilde{T}h_i(T) \log(T)}$$

$$\leq \frac{(\sqrt{\alpha_{\max}} + \sqrt{\beta})^2 \sqrt{(M - K^* + 1)}}{(B_{K^*} - B_i)} \log(T).$$

Let $j_{\max}(t) = \arg\max_{j\in[M]}\left((S_j - S_{K^*}) + (\sqrt{\alpha_{s,j}} + \sqrt{\beta})\sqrt{\frac{\log(t)}{n_{b,j}(t)}}\right)$. We have under the event $\mathcal{E}^{(\beta)} = \cup_{t=1}^T \mathcal{E}_t^{(\beta)}$ and $\alpha_{\min} > \beta$,

$$\sum_t \left(\max_{j\in\mathcal{P}_s(t)} s_j(t) - S_{K^*}\right)$$

$$\leq \sum_t \max_{j\in\mathcal{P}_s(t)} \left((S_j - S_{K^*}) + (\sqrt{\alpha_{max}} + \sqrt{\beta})\sqrt{\frac{\log(t)}{n_{b,j}(t)}}\right)$$

$$\leq \sum_{j\in[M]} \sum_{t:\chi_{s,j}(t)=1,j_{\max}(t)=j} \left((S_j - S_{K^*}) + (\sqrt{\alpha_{max}} + \sqrt{\beta})\sqrt{\frac{\log(t)}{n_{b,j}(t)}}\right)$$

Further, under the event $\mathcal{E}^{(\beta)}$ and $\alpha_{\min} > \beta$, we know that $K(t) \geq K^*$, which implies at least one seller $j \geq K^*$ is active. Hence, for any $j < K^*$, $j_{\max}(t) = j$ only if

$$n_{b,j}(t) \leq \tilde{T}h_j(T) = \min_{j'\geq K^*} \frac{(\sqrt{\alpha_{max}} + \sqrt{\beta})^2}{(S_{j'} - S_j)^2} \log(T) = \frac{(\sqrt{\alpha_{max}} + \sqrt{\beta})^2}{(S_{K^*} - S_j)^2} \log(T).$$

Also, the maximum number of times any seller $j \geq (K^* + 1)$ participates under the event $\mathcal{E}^{(\beta)}$ and $\alpha_{\min} > \beta$ is $Th_j(T) \geq \frac{(N-K^*+1)(\sqrt{\alpha_{max}}+\sqrt{\beta})^2}{(S_j - S_{K^*})^2} \log(T)$, according to 13.

Therefore, we can proceed as

$$\sum_{t:\chi_{b,\tilde{i}}(t)=1} (\max_{j\in\mathcal{P}_s(t)} s_j(t) - S_{K^*})$$

$$\leq \sum_{j\in[M]} \sum_{t:\chi_{s,j}(t)\chi_{s,\tilde{i}}(t)=1,j_{\max}(t)=j} \left((S_j - S_{K^*}) + (\sqrt{\alpha_{max}} + \sqrt{\beta})\sqrt{\frac{\log(t)}{n_{b,j}(t)}}\right)$$

$$\leq \sum_{j\geq(K^*+1)} \sum_{t:\chi_{s,j}(t)=1} \left((S_j - S_{K^*}) + (\sqrt{\alpha_{max}} + \sqrt{\beta})\sqrt{\frac{\log(t)}{n_{b,j}(t)}}\right)$$

$$+ \sum_{j\leq K^*} \sum_{t:\chi_{s,j}(t)\chi_{b,\tilde{i}}(t)=1,j_{\max}(t)=j} \left((S_j - S_{K^*}) + (\sqrt{\alpha_{max}} + \sqrt{\beta})\sqrt{\frac{\log(t)}{n_{b,j}(t)}}\right)$$

$$\leq \sum_{j\geq(K^*+1)} \left(Th_j(T)(S_j - S_{K^*}) + \sum_{n=1}^{Th_j(T)} (\sqrt{\alpha_{max}} + \sqrt{\beta})\sqrt{\frac{\log(T)}{n}}\right)$$

$$+ \sum_{j<K^*} \sum_{n=1}^{\tilde{T}h_j(T)} (\sqrt{\alpha_{max}} + \sqrt{\beta})\sqrt{\frac{\log(T)}{n}} + \sum_{n=1}^{n_{b,\tilde{i}}(T)} (\sqrt{\alpha_{max}} + \sqrt{\beta})\sqrt{\frac{\log(T)}{n}}$$

$$\leq \sum_{j\geq(K^*+1)} \left(Th_j(T)(S_j - S_{K^*}) + (\sqrt{\alpha_{max}} + \sqrt{\beta})\sqrt{Th_j(T)\log(T)}\right)$$

$$+ \sum_{j<K^*} (\sqrt{\alpha_{max}} + \sqrt{\beta})\sqrt{\tilde{T}h_j(T)\log(T)} + (\sqrt{\alpha_{max}} + \sqrt{\beta})\sqrt{n_{b,\tilde{i}}(T)\log(T)}$$

$$\leq \sum_{j\geq(K^*+1)} \left(\frac{(N-K^*+1)(\sqrt{\alpha_{max}}+\sqrt{\beta})^2}{(S_j - S_{K^*})} + \frac{\sqrt{(N-K^*+1)}(\sqrt{\alpha_{max}}+\sqrt{\beta})^2}{(S_j - S_{K^*})}\right)\log(T)$$

$$+ \sum_{j<K^*} \frac{(\sqrt{\alpha_{max}}+\sqrt{\beta})^2}{(S_{K^*} - S_j)} \log(T) + (\sqrt{\alpha_{max}} + \sqrt{\beta})\sqrt{n_{b,\tilde{i}}(T)\log(T)}$$

Combining the above bounds, we get

$$2\sum_{t:\chi_{b,\tilde{i}}(t)=1} (p(t) - p^*)$$

$$\leq \underbrace{2(\sqrt{\alpha_{max}} + \sqrt{\beta})\sqrt{n_{b,\tilde{i}}(T)\log(T)}}_{\text{price setting buyer and seller}} + \underbrace{\sum_{j<K^*} \frac{(\sqrt{\alpha_{max}} + \sqrt{\beta})^2}{(S_{K^*} - S_j)}\log(T)}_{\text{non-price setting participant sellers}}$$

$$+ \underbrace{\sum_{i \geq (K^*+1)} \frac{(\sqrt{\alpha_{max}} + \sqrt{\beta})^2\sqrt{(M - K^* + 1)}}{(B_{K^*} - B_i)}\log(T)}_{\text{non-participating buyers}}$$

$$+ \underbrace{\sum_{j \geq (K^*+1)} \left( \frac{(N - K^* + 1)(\sqrt{\alpha_{max}} + \sqrt{\beta})^2}{(S_j - S_{K^*})} + \frac{\sqrt{(N - K^* + 1)}(\sqrt{\alpha_{max}} + \sqrt{\beta})^2}{(S_j - S_{K^*})} \right)\log(T)}_{\text{non-participating sellers}}.$$

Reversing the role of buyer and seller in the above derivation, and leveraging the symmetry in the system, we can get

$$2\sum_{t:\chi_{s,\tilde{j}}(t)=1} (p^* - p(t))$$

$$\leq \underbrace{2(\sqrt{\alpha_{max}} + \sqrt{\beta})\sqrt{n_{s,\tilde{j}}(T)\log(T)}}_{\text{price setting buyer and seller}} + \underbrace{\sum_{i<K^*} \frac{(\sqrt{\alpha_{max}} + \sqrt{\beta})^2}{(B_i - B_{K^*})}\log(T)}_{\text{non-price setting participant buyers}}$$

$$+ \underbrace{\sum_{j \geq (K^*+1)} \frac{(\sqrt{\alpha_{max}} + \sqrt{\beta})^2\sqrt{(N - K^* + 1)}}{(S_j - S_{K^*})}\log(T)}_{\text{non-participating sellers}}$$

$$+ \underbrace{\sum_{i \geq (K^*+1)} \left( \frac{(M - K^* + 1)(\sqrt{\alpha_{max}} + \sqrt{\beta})^2}{(B_{K^*} - B_i)} + \frac{\sqrt{(M - K^* + 1)}(\sqrt{\alpha_{max}} + \sqrt{\beta})^2}{(B_{K^*} - B_i)} \right)\log(T)}_{\text{non-participating buyers}}.$$

$\square$

# B   Proofs of the Lower Bounds

## B.1   Minimax lower bound on individual regret

We show a minimax regret lower bound of $\Omega(\sqrt{T})$ in Lemma 21 by considering a simpler system that decouples learning and competition. In this system, the seller is assumed to *(i)* know her exact valuation, and *(ii)* always ask her true valuation as the selling price, i.e., is truthful in her asking price in all the rounds. Furthermore, the pricing at every round is fixed to the average $p_t = \frac{B_t + S_t}{2}$ in the event that $B_t \geq S$. The utility of the buyer at time $t$ is defined as $U_t = (p_t - B)\mathbf{1}(B_t \geq S)$.

**Lemma 18.** *The utility maximizing action of an oracle buyer that knows $B$ and $S$ is to bid $B_t = S\mathbf{1}(B \geq S) + (S - \varepsilon)\mathbf{1}(B < S)$ for any $\varepsilon > 0$, at all times. In words, the oracle buyer either bids $S$ and pays the price $S$ or "abstains" by bidding less than $S$.*

**Corollary 19.** *To minimize expected utility in the simple system, it suffices for the buyer each round to decide to either bid $S$ and participate in the market by paying price $S$, or abstain without participation and obtain no reward.*

**Reduction to a two armed bandit problem:** Corollary 19 gives that at each time, it suffices for the buyer that does not know her true valuation to either bid $S$ and participate at price $S$, or abstain from participating. In any round $t$ that the buyer participates, she obtains a mean reward of $B - S$, while she receives $0$ reward in rounds she abstains from participating. Thus, the actions of the buyer are equivalent to a two armed bandit, one with mean $B - S$ and the other is deterministic $0$ mean. The reduction is formalized in the following Corollary.

**Corollary 20.** *Any bidding policy given in Definition 24 describes a two-armed bandit policy (Definition 25) with arm-means $B - S$ and $0$.*

**Lemma 21.** *For every bidding policy, there exists a system such that $\mathbb{E}[R_T] \geq \frac{1}{36}\sqrt{T}$.*

The proof follows from [28] and is reproduced in Appendix B.4 for completeness. Further, we show in Appendix B.6 the lower bound can be extended to a system of multiple buyers and sellers. Thus, our upper bound of $O(\sqrt{T})$ is order-wise optimal as Lemma 21 and Corollary 20 show that $O(\sqrt{T})$ regret bound is un-avoidable even in the absence of competition.

## B.2 Lower bound on Social Welfare Regret

The key observation is that social-welfare regret in Equation (1) is *independent* of the pricing mechanism and only depends on the participating buyers $\mathcal{P}_b(t)$ and sellers $\mathcal{P}_s(t)$ at each time $t$. We will establish a lower bound on a centralized decision maker (DM), who at each time, observes all the rewards obtained by all agents thus far, and decides $\mathcal{P}_b(t)$ and $\mathcal{P}_s(t)$ for each $t$. In the Appendix in Section B.7, we show that the actions of the DM can be coupled to that of a combinatorial semi-bandit model [14], where the base set of arms are the set of all buyers and sellers $\{B_1, \cdots, B_M\} \cup \{S_1, \cdots, S_N\}$ is the set of buyers and sellers, the valid subset of arms are those that have an equal number of buyers and sellers and the mean reward of any valid subset $\mathcal{A} \subseteq 2^{\mathcal{D}}$ is the difference between the sum of all valuations of buyers in $\mathcal{A}$ and of sellers in $\mathcal{A}$.

**Proposition 22.** *The optimal action for the centralized DM is to pick $\mathcal{P}_b^* \cup \mathcal{P}_s^*$.*

Proof is deferred to Section B.7. Thus, the regret of the centralized decision maker is $R_{\text{SB}} = T(\sum_{i \in \mathcal{P}_b^*} B_i - \sum_{j \in \mathcal{P}_s^*} S_j) - \sum_{t=1}^{T}(\sum_{i \in \mathcal{P}_b(t)} B_i - \sum_{j \in \mathcal{P}_s(t)} S_j)$. By by adding and subtracting $\sum_{j \in [M]} S_j$ to both sides, we get that $R_{\text{SB}}$ is identical to $R_{\text{SW}}$ given in Equation (1). Thus, a lower bound on $R_{SB}$ implies a lower bound on $R_{SW}$.

**Lemma 23** (Theorem 1 [14])**.** *Suppose the reward distributions for all agents (buyers and sellers) are unit variance gaussians. Then, the regret of any uniformly good policy[6] for the combinatorial semi-bandit suffers regret $\limsup_{T \to \infty} \frac{R_T}{\log(T)} = c((\mu_a)_{a \in \mathcal{D}})$, where the constant $c((\mu_a)_{a \in \mathcal{D}}) > 0$ is strictly positive if the best and the next best subsets have different mean rewards.*

A restatement and proof is given in Appendix B.7. Thus our upper bound of $O(\log(T))$ social welfare regret under the decentralized setting is order optimal since even a centralized system must incur $\Omega(\log(T))$ regret.

## B.3 Proof of Lemma 18

*Proof.* Assume the oracle buyer knows $B, S$, the average price mechanism and the fact that the seller is not strategizing.

**Case I :** $B < S$. If the buyer bids $B_t \geq S$, then the buyer will be matched with price $p_t \geq S$ and will receive an utility $U_t := B - p_t < 0$. If on the other hand, the buyer puts any bid $B_t < S$, then the buyer is not matched and receives an utility of $0$. Thus, the optimal choice for the oracle buyer in this case is to place any bid $B_t < S$. Thus, bidding $B_t := S - \varepsilon$, for every $\varepsilon > 0$ is optimal.

**Case II :** $B \geq S$. If the buyer bids $B_t \geq S$, then the buyer will be matched with price $p_t = \frac{B_t + S}{2}$ and will receive an utility $U_t := B - p_t = \frac{2B - B_t - S}{2}$. Observe that the utility $U_t$ is non-decreasing in the bid-price $B_t$. If $B_t < S$ however, no match occurs and the oracle buyer will receive $0$ utility. If on the other-hand $S \leq B_t \leq B$, then $B - B_t \geq 0$ and $B - S \geq 0$. Thus, the utility $U_t \geq 0$. This along with the non-increasing nature of $U_t$ gives that the optimal action is to play $B_t = S$. $\qquad \square$

## B.4 Proof of Lemma 21

*Proof.* This follows the same recipe of bandit lower bounds [28]. Let the bidding policy be arbitrary as in the hypothesis of the lemma. Fix a system and denote by $\varepsilon := S - B$. We will choose an

---

[6]This is a standard technical condition defined in the Appendix in Section B.8

appropriate value of $\varepsilon$ later in the proof. Denote by this system where $S - B = \varepsilon$ as $\nu$. Denote by the system in which $B = S + \varepsilon$ as $\nu'$. We denote by $R_T$ and $R'_T$ to be the regret obtained by the bidding policy in system $\nu$ and $\nu'$ respectively. The first observation we will make is that the divergence decomposition lemma gives

$$\mathbb{E}_\nu \left[ \sum_{t=1}^{T} Z_t \right] D(-\varepsilon, \varepsilon) \geq \log \left( \frac{1}{2(\mathbb{P}_\nu(A) + \mathbb{P}_{\nu'}(A^\complement))} \right), \tag{5}$$

for any measurable event $A$. The proof of this claim follows from the well known Bretagnolle–Huber inequality (Theorem 14.2 [28]) and the divergence decomposition lemma (Lemma 15.1 [28]) to compute the divergence between the system $\nu$ and $\nu'$. The formula from Lemma 15.1 of [28] when applied to our system simplifies to the LHS of Equation (5) by observing the fact that in both system $\nu$ and system $\nu'$, not participating in the market gives a deterministic $0$ reward. Thus, the KL divergence between the reward distributions for not participating is $0$.

The rest of the proof is to verbatim follow the proof of Theorem 15.2 of [28] to re-arrange Equation (5) to yield the desired result. We denote by the event $A := \left\{ \sum_{t=1}^{T} Z_t \geq T/2 \right\}$. Thus, trivially, $R_T \geq \frac{\varepsilon T}{2} \mathbb{P}_\nu[A]$ and $R'_T \geq \frac{\varepsilon T}{2} \mathbb{P}_\nu[A^\complement]$. Furthermore, $\mathbb{E} \left[ \sum_{t=1}^{T} Z_t \right] \leq T$. Now, using the fact that for unit variance gaussians, $D(-\varepsilon, \varepsilon) = 2\varepsilon^2$ and plugging these estimates in Equation (5), we obtain that

$$R_T + R'_T \geq \frac{\varepsilon T}{4} \exp \left( -2\varepsilon^2 T \right).$$

As $\varepsilon > 0$ was arbitrary, we can set it to be equal to $\frac{1}{\sqrt{T}}$, and use the well known fact that for any non-negative $a, b$, we have $a + b \leq 2 \max(a, b)$, to get

$$\max(R_T, R'_T) \geq \frac{1}{36} \sqrt{T}.$$

$\square$

### B.5 Proof of Corollary 19

**Definition 24** (Bidding Policy)**.** *A sequence of binary random variables $(Z_t)_{t \geq 1}$, such that for all $t \geq 1$, $Z_t \in \{0, 1\}$ denotes whether the buyer participates (by placing a bid of $S$) in the tth round or not such that $Z_t$ is measurable with respect to the sigma-algebra generated by decisions and rewards $Z_1 Y_{b,1}, \cdots, Z_{t-1} Y_{b,t-1}$ observed till time $t - 1$.*

**Definition 25** (Two-armed bandit policy:)**.** *A sequence of $\{0, 1\}$ valued random variables $(\widehat{Z}_t)_{t \geq 1}$ such that for all time $t$, $\widehat{Z}_t$ is measurable with respect to a sequence of $\mathbb{R}$ valued random variables $\mathcal{F}_{t-1} := \sigma(\widehat{X}_1, \cdots, \widehat{X}_{t-1})$ such that for all time $t$, conditioned on $\widehat{Z}_t$, (i) $\widehat{X}_t$ is independent of $\mathcal{F}_{t-1}$, and (ii) $\widehat{X}_t \sim \mathcal{P}_{\widehat{Z}_t}$, where $\mathcal{P}_i$, $i \in \{1, 2\}$ are two fixed probability distributions on $\mathbb{R}$.*

This definition formalizes the intuitive notion that a sequence of binary decisions made at each time, with the decision being measurable function of all past observations and the observations themselves being independent conditioned on the arm chosen.

*Proof.* A sequence of binary valued random variables $\{Z_\cdot, \cdots, Z_T\}$ and $\mathbb{R}$ valued observation random variables $\{Y_1, \cdots, Y_T\}$ describes a policy for a two armed stochastic bandits if and only if *(i)* $Z_t$ is measurable with respect to the sigma-algebra $\mathcal{F}_{t-1} := \sigma(Y_1, \cdots, Y_{t-1})$ generated by all observed rewards upto time $t - 1$, and *(ii)* conditioned on the arm $Z_t$, the observed reward $Y_t$ is conditionally independent of $(Y_1, \cdots, Y_{t-1})$ and $(Z_1, \cdots, Z_{t-1})$ the actions and rewards obtained in the past. It is easy to observe both of these for the bidding system described before. $\square$

### B.6 Extension to the multi-agent setting

The simplified system in 5.1 specified a single buyer and seller system that had no competition as the seller's behaviour was fixed. In this section, we show that a $\sqrt{T}$ minimax lower bound is inevitable even in an appropriately simplified multi-agent system that decouples learning from competition.

In this simplified setting, we assume *(i)* the selling price $p_t := p^*$ is set constant for every round $t$, and *(ii)* there is no shortage of goods, i.e., any buyer(seller) that wants to participate in a given round by paying(selling at) $p^*$, can buy(sell) so. Thus, under this setup, the entire market is a decoupled union of individual agents interacting against a fixed environment dictated by a price $p^*$. It can be observed, that for any buyer(seller) in this simplified setting, the lower bound from Lemma 21 applies verbatim following the same coupling arguments.

**Corollary 26.** *For every bidding policy for the general system, for any seller $j \in [M]$ there exists a system such that $\mathbb{E}[R_{s,j}(T)] \geq \frac{1}{36}\sqrt{T}$. Similarly, for any buyer $i \in [N]$ there exists a system such that $\mathbb{E}[R_{b,i}(T)] \geq \frac{1}{36}\sqrt{T}$.*

## B.7 Reduction of the centralized DM in Section B.2 to a combinatorial semi-bandit model

A combinatorial semi-bandit [11] is the following variant of the standard $\mathcal{D} := \{1, \cdots, D\}$ armed bandit problem. At each time $t$, the decision maker chooses a subset $\mathcal{A}_t \subseteq \mathcal{D}$ from a fixed set of subsets $\boldsymbol{\mathcal{A}} \subseteq 2^{\mathcal{D}}$. At each time $t$, the received reward is $\sum_{a \in \mathcal{A}_t} X_t(a)$, where for every arm $a \in \mathcal{D}$, $(X_t(a))_{t \geq 1}$ is an i.i.d. sequence with mean $\mu_a$. Later in Lemma 27, we will assume that the reward distriutions are unit-variance gaussians. The arm-means $(\mu_a)_{a \in \mathcal{D}}$ are a-priori unknown to the decision maker. The goal of the decision maker is to minimize the cumulative regret defined as $R_{SB} := \max_{\mathcal{A} \in \boldsymbol{\mathcal{A}}} T \sum_{a \in \mathcal{A}} \mu_a - \sum_{t=1}^{T} \sum_{a \in \mathcal{A}_t} \mu_a$.

Consider the centralized DM who at each time decides the participating buyers and sellers $\mathcal{P}_b(t)$ and $\mathcal{P}_s(t)$. The base set of 'arms' $\mathcal{D} := \{B_1, \cdots, B_N\} \cup \{S_1, \cdots, S_M\}$ is the set of buyers and sellers. The mean reward $\mu_a$ for any buyer $a$ is their unknown valuation, while $\mu_a$ for any seller $a$ is the *negative* of their true valuation. The set of allowed subsets $\boldsymbol{\mathcal{A}}$ are those subsets of $\mathcal{D}$ that contain the same number of buyers and sellers.

### B.7.1 Proof of Proposition 22

*Proof.* The proof follows from two observations. *(i)* the mean reward for any buyer is positive and that for the seller is negative. *(ii)* the set of allowed subsets to play consists of an equal number of buyers and sellers. Thus, from definition of $\mathcal{P}_b^*$ and $\mathcal{P}_s^*$, every other subset satisfying condition *(ii)* will not have mean reward strictly larger than that of $\mathcal{P}_b^* \cup \mathcal{P}_s^*$. $\qquad\square$

Thus, the regret of the centralized decision maker is $R_{\text{SB}} = T(\sum_{i \in \mathcal{P}_b^*} B_i - \sum_{j \in \mathcal{P}_s^*} S_j) - \sum_{t=1}^{T}(\sum_{i \in \mathcal{P}_b(t)} B_i - \sum_{j \in \mathcal{P}_s(t)} S_j)$. By by adding and subtracting $\sum_{j \in [M]} S_j$ to both sides, we get that

$$R_{\text{SB}} = T(\sum_{i \in \mathcal{P}_b^*} B_i + \sum_{j \in [M] \setminus \mathcal{P}_s^*} S_j) - \sum_{t=1}^{T}(\sum_{i \in \mathcal{P}_b(t)} B_i + \sum_{j \in [M] \setminus \mathcal{P}_s(t)} S_j). \tag{6}$$

As Equation (6) is identical to the definition of the social regret in Equation (1), a lower bound on $R_{SB}$ implies a lower bound on the social-welfare regret $R_{SW}$.

**Lemma 27** (Theorem 1 [14])**.** *Suppose the reward distributions for all agents (buyers and sellers) are unit variance gaussians. Then, the regret of any uniformly good policy[7] for the combinatorial semi-bandit suffers regret $\limsup_{T \to \infty} \frac{R_T}{\log(T)} = c(\boldsymbol{\mathcal{A}}; (\mu_a)_{a \in \mathcal{D}})$, where the constant $c(\boldsymbol{\mathcal{A}}; (\mu_a)_{a \in \mathcal{D}}) > 0$ is strictly positive if the best and the next best subsets have different mean rewards.*

Thus our upper bound of $O(\log(T))$ social welfare regret under the decentralized setting is order optimal since even a centralized system must incur $\Omega(\log(T))$ regret.

## B.8 Additional Details on Combinatorial Semi-Bandits

For completeness, we collect all relevant definitions and statements from [14] that are used in the lower bound proof in Section B.2. Recall the setup – at each time $t$, the decision maker can choose a subset of arms from a given set $\boldsymbol{\mathcal{A}} \subseteq 2^{\mathcal{D}}$ of subsets. In this setting, the mean reward obtained by any arm is a unit variance gaussian random variable. Let $\Theta$ denote the set of possible parameters for the means which in

---

[7]This is a standard technical condition defined in the Appendix in Section B.8

our example is the set of $M + N$ means, i.e., $\Theta := \mathbb{R}^{M+N}$, and $\theta \in \Theta$ denote the mean reward for the elements in $\mathcal{D}$. For any subset $\mathcal{A} \in \pmb{\mathcal{A}}$, let $\mu_{\mathcal{A}}(\theta) := \sum_{a \in \mathcal{A}} \theta_a$. We use $\mu_{\mathcal{A}}$ instead of $\mu_{\mathcal{A}}(\theta)$ when the underlying instance $\theta$ is clear. Let the subset with highest mean be $\mathcal{A}^*(\theta) := \arg\max_{\mathcal{A} \in \pmb{\mathcal{A}}} \sum_{a \in \mathcal{A}} \theta_a$, and $\mu^*(\theta)$ be the highest reward, i.e., $\mu^*(\theta) = \max_{\mathcal{A} \in \pmb{\mathcal{A}}} \mu_{\mathcal{A}}(\theta) := \mu_{\mathcal{A}^*(\theta)}(\theta)$. An assumption we make throughout in Section B.2 is that $\mathcal{A}^*$ is unique and the gap is strictly positive, i.e.

$$\Delta(\pmb{\mathcal{A}}; \theta) := \min_{\mathcal{A} \in \pmb{\mathcal{A}} \setminus \{\mathcal{A}^*\}} (\mu^*(\theta) - \mu_{\mathcal{A}}(\theta)) > 0. \tag{7}$$

A policy $\pi$ for the decision maker is a measurable function that at each time $t$, maps all the observed rewards of the arms pulled to a subset $\mathcal{A}_t^\pi$ to play. Let $R^\pi(T)$ be the $T$ round regret of the policy $\pi$ for an instance $\theta \in \Theta$, which is defined as $R^\pi(T; \theta) = \sum_{t=1}^{T} (\mu_{\mathcal{A}^*}(\theta) - \mu_{\mathcal{A}_t^\pi}(\theta))$.

**Definition 28** (Uniformly Good). *A policy $\pi$ for the DM is uniformly good, if for all problem settings $\theta \in \Theta$ satisfying assumption in Equation (7), $R^\pi(T; \theta) = o(T^\alpha)$, for every $\alpha \in (0, 1)$.*

**Theorem 29** (Theorem 1 in [14]). *For any instance $\theta \in \Theta$ satisfying the assumption in Equation (7) and any uniformly good policy $\pi$, $\limsup_{T \to \infty} \frac{R^\pi(T;\theta)}{\log(T)} \geq c(\pmb{\mathcal{A}}; \theta)$, where $c(\pmb{\mathcal{A}}; \theta)$ is given by the solution of the following optimization problem*

$$c(\pmb{\mathcal{A}}; \theta) := \inf_{x \in \mathbb{R}_+^{|\pmb{\mathcal{A}}|}} \sum_{\mathcal{A} \in \pmb{\mathcal{A}}} x_{\mathcal{A}} (\mu^*(\theta) - \mu_{\mathcal{A}}(\theta)) \text{ such that}$$

$$\sum_{a \in \mathcal{D}} \left( \sum_{\mathcal{A} \in \pmb{\mathcal{A}} \text{ s.t. } a \in \mathcal{A}} x_{\mathcal{A}} \right) kl(\theta_a, \lambda_a) \geq 1, \ \forall \lambda \in B(\theta),$$

*where $B(\theta) := \{\lambda \in \Theta, \text{ s.t. } \theta_i = \lambda_i \ \forall i \in \mathcal{A}^*(\theta), \mu^*(\lambda) > \mu^*(\theta)\}$, and for any $u, v \in \mathbb{R}$, $kl(u, v) = \frac{1}{2}(u - v)^2$.*

The proof of this theorem is given as the proof of theorem 1 in [14]. Although the proof in [14] is given in the case for Bernoulli distributed arms, the proof follows verbatim for gaussian distributed arms by using the fact that the KL divergence between two unit variance gaussians with means $u, v \in \mathbb{R}$ is $kl(u, v) := \frac{1}{2}(u - v)^2$.

We give a simple lower bound to $c(\pmb{\mathcal{A}}; \theta)$ in the following lemma.

**Lemma 30.**

$$c(\mathcal{A}; \theta) \geq \sum_{i=K^*+1}^{N} \frac{2}{\min(B_{K^*}, S_{K^*+1}) - B_i} + \sum_{j=K^*+1}^{M} \frac{2}{S_j - \max(S_{K^*}, B_{K^*+1})}$$

*Proof.* For ease of exposition, we will use the notation that for any buyer $i \in \{1, \cdots, N\}$, $\theta_{b,i}$ to be the true valuation of buyer $i$ under model $\theta$. Similarly, for any seller $j \in \{1, \cdots, M\}$, we denote by $\theta_{s,j}$ to be the true valuation of seller $j$ under model $\theta$. Observe that under the mapping to the combinatorial bandits, we denote by all agents' indices by the set of cardinality $M + N$, denoted by $\mathcal{D}$. We denote by the first $N$ indices of $\mathcal{D}$ to represent the buyers and the last $M$ indices of $\mathcal{D}$ to denote the sellers.

For every $\varepsilon > 0$, let

$B_0(\theta) = \{\lambda \text{ s.t. }, \forall i, j \leq K^* \lambda_{b,i} = \theta_{b,i}, \lambda_{s,j} = \theta_{s,j}\}$,

$B_1(\theta; \varepsilon) = \{\lambda \in B_0(\theta); \exists \text{ exactly one } i' \in \{K^* + 1, \ldots, N\}, \lambda_{b,i'} = \min(\theta_{b,K^*}, \theta_{s,K^*+1}) + \varepsilon\}$,

$B_2(\theta; \varepsilon) = \{\lambda \in B_0(\theta); \exists \text{ exactly one } j' \in \{K^* + 1, \ldots, M\}, \lambda_{s,j'} = \max(\theta_{s,K^*}, \theta_{b,K^*+1}) - \varepsilon\}$.

Observe by definition that for all $\varepsilon > 0$, $B_1(\theta, \varepsilon) \cap B_2(\theta, \varepsilon) = \emptyset$ and $B_1(\theta, \varepsilon) \cup B_2(\theta, \varepsilon) \subset B(\theta)$. Define by $c_\varepsilon(\pmb{\mathcal{A}}; \theta)$ to be the solution to the following optimization problem

$$c_\varepsilon(\pmb{\mathcal{A}}; \theta) := \inf_{x \in \mathbb{R}_+^{|\pmb{\mathcal{A}}|}} \sum_{\mathcal{A} \in \pmb{\mathcal{A}}} x_{\mathcal{A}} (\mu^*(\theta) - \mu_{\mathcal{A}}(\theta)) \text{ such that} \tag{8}$$

$$\sum_{a \in \mathcal{D}} \left( \sum_{\mathcal{A} \in \pmb{\mathcal{A}} \text{ s.t. } a \in \mathcal{A}} x_{\mathcal{A}} \right) kl(\theta_a, \lambda_a) \geq 1, \ \forall \lambda \in \mathcal{B}_1(\theta, \varepsilon) \cup B_2(\theta, \varepsilon).$$

The optimization problem for $c_\varepsilon(\mathcal{A}; \theta)$ differs from that for $c(\mathcal{A}; \theta)$ in Theorem 29 in the constraints. Since $B_1(\theta, \varepsilon) \cup B_2(\theta, \varepsilon) \subset B(\theta)$, we have that $c(\mathcal{A}; \theta) \geq c_\varepsilon(\mathcal{A}; \theta)$. As this in-equality holds for all $\varepsilon > 0$, we have that

$$c(\mathcal{A}; \theta) \geq \sup_{\varepsilon > 0} c_\varepsilon(\mathcal{A}; \theta).$$

In the rest of this proof, we will develop a lower bound for $\sup_{\varepsilon > 0} c_\varepsilon(\mathcal{A}; \theta)$.

For notational simplicity, we denote by $y_a := \sum_{\mathcal{A} \in \mathcal{A}: a \in \mathcal{A}} x_{\mathcal{A}}$. The constraint in the optimization problem in Equation (8) yields that, for all buyers $i' \in \{K^* + 1, \cdots, N\}$ and all sellers $j' \in \{K^* + 1, \cdots, M\}$, we need to have

$$y_{b,i'} \geq \frac{1}{kl(\theta_{b,i'}, \min(\theta_{b,K^*}, \theta_{s,K^*+1}) + \varepsilon)}, \text{ and } y_{s,j'} \geq \frac{1}{kl(\theta_{s,j'}, \max(\theta_{b,K^*+1}, \theta_{s,K^*}) - \varepsilon)}.$$

Additionally, monotonicity yields that for any buyer $i' \in \{K^* + 1, \cdots, N\}$,

$$\mathcal{A}_b^{(i')} := \arg\min_{\mathcal{A} \in \mathcal{A}: B_{i'} \in \mathcal{A}} (\mu^*(\theta) - \mu_{\mathcal{A}}(\theta)) = \begin{cases} \{B_1, \ldots, B_{(K^*-1)}, B_{i'}\} \cup \{S_1, \ldots, S_{K^*}\}, & B_{K^*} < S_{K^*+1}, \\ \{B_1, \ldots, B_{K^*}, B_{i'}\} \cup \{S_1, \ldots, S_{K^*}, S_{K^*+1}\}, & B_{K^*} > S_{K^*+1}. \end{cases}$$

Similarly, for any seller $j' \in \{K^* + 1, \cdots, N\}$,

$$\mathcal{A}_s^{(j')} := \arg\min_{\mathcal{A} \in \mathcal{A}: S_{j'} \in \mathcal{A}} (\mu^*(\theta) - \mu_{\mathcal{A}}(\theta)) = \begin{cases} \{B_1, \ldots, B_{K^*}\} \cup \{S_1, \ldots, S_{(K^*-1)}, S_{j'}\}, & B_{K^*+1} < S_{K^*}, \\ \{B_1, \ldots, B_{K^*}, B_{K^*+1}\} \cup \{S_1, \ldots, S_{K^*}, S_{j'}\}, & B_{K^*+1} > S_{K^*}. \end{cases}$$

Note that for all $i', j', i'', j'' \geq (K^* + 1)$, $i' \neq i''$ and $j' \neq j''$, $B_{i''} \notin \mathcal{A}_b^{(i')}$, and $S_{j''} \notin \mathcal{A}_s^{(j')}$. Therefore, the minimizer of Equation (8) is given by assigning

$$\forall i' \in \{K^* + 1, \cdots, N\}, \; x_{\mathcal{A}_b^{(i')}} := \frac{1}{kl(\theta_{b,i'}, \min(\theta_{b,K^*}, \theta_{s,K^*+1}) + \varepsilon)},$$

$$\forall j' \in \{K^* + 1, \ldots, M\}, \; x_{\mathcal{A}_s^{(j')}} := \frac{1}{kl(\theta_{s,j'}, \max(\theta_{b,K^*+1}, \theta_{s,K^*}) - \varepsilon)},$$

and for all other sets $\mathcal{A}, x_{\mathcal{A}} = 0$.

This ensures for all buyers $i' \in \{K^* + 1, \cdots, N\}$ and all sellers $j' \in \{K^* + 1, \cdots, M\}$,

$$y_{b,i'} = \frac{1}{kl(\theta_{b,i'}, \min(\theta_{b,K^*}, \theta_{s,K^*+1}) + \varepsilon)}, \text{ and } y_{s,j'} = \frac{1}{kl(\theta_{s,j'}, \max(\theta_{b,K^*+1}, \theta_{s,K^*}) - \varepsilon)}.$$

Thus, we get

$$c_\varepsilon(\mathcal{A}; \theta) = \sum_{i'=K^*+1}^{N} \frac{\mu^*(\theta) - \mu_{\mathcal{A}_b^{(i')}}(\theta)}{kl(\theta_{b,i'}, \min(\theta_{b,K^*}, \theta_{s,K^*+1}) + \varepsilon)} + \sum_{j'=K^*+1}^{M} \frac{\mu^*(\theta) - \mu_{\mathcal{A}_s^{(j')}}(\theta)}{kl(\theta_{s,j'}, \max(\theta_{b,K^*+1}, \theta_{s,K^*}) - \varepsilon)},$$

$$= \sum_{i'=K^*+1}^{N} \frac{\min(\theta_{b,K^*}, \theta_{s,K^*+1}) + \varepsilon - \theta_{b,i'}}{kl(\theta_{b,i'}, \min(\theta_{b,K^*}, \theta_{s,K^*+1}) + \varepsilon)} + \sum_{j'=K^*+1}^{M} \frac{\theta_{s,j'} - \max(\theta_{s,K^*}, \theta_{b,K^*+1}) + \varepsilon}{kl(\theta_{s,j'}, \max(\theta_{b,K^*+1}, \theta_{s,K^*}) - \varepsilon)}$$

Optimizing over $\varepsilon > 0$ and using the fact that $kl(u, v) := \frac{1}{2}(u - v)^2$ yields the result. $\square$

## C  Deviations and Incentives of Agents

We now discuss how agents may deviate under average mechanism under their own incentive. We limit ourselves to deviations from *myopic oracle* agents, each of whom optimizes her single round reward, and possess an oracle knowledge of her own valuation. Our incentives are shaped by symmetric equilibrium in double auction [42], where all non-strategic buyers employ same strategy, and the same hold for non-strategic sellers. In particular, all the non-strategic agents use confidence-based

bidding. The strategic deviant agents do not deviate from a strategy if incremental deviations in bids do not improve their own reward.

Under the above setup, for average mechanism only the price-setting agents (i.e. the $K^*$-th buyer and $K^*$-th seller) have incentive to deviate from their true valuation to increase their single-round reward (c.f. Section 5 in [42]). For other agents, deviations from reporting their true value does not improve their instantaneous reward. Thus average mechanism is *truthful for non-price-setting agents*.

For the price-setting agents, the incentives, and impact on regret are as follows.

1. *Only $K^*$-th buyer deviates:* The $K^*$-th buyer has an incentive to set her bid *close*[8] to $\max(S_{K^*}, B_{K^*+1})$. The long term average price is now set close to $(S_{K^*} + \max(S_{K^*}, B_{K^*+1}))/2$. When compared to Average mechanism outcomes this leads to $(B_{K^*} - \max(S_{K^*}, B_{K^*+1}))/2$ average surplus in each round for participating buyers, and the same average deficit in each round for participating sellers.

2. *Only $K^*$-th seller deviates:* The $K^*$-th seller has an incentive to set her bid close to $\min(B_{K^*}, S_{K^*+1})$. With a long term average price of $(B_{K^*} + \min(B_{K^*}, S_{K^*+1}))/2$, each seller has a per round $(S_{K^*} - \min(B_{K^*}, S_{K^*+1}))/2$ surplus, and each buyer has the same average deficit in each round.

3. *Both $K^*$-th seller and buyer deviate:* The $K^*$-th seller has an incentive to set her bid close to $\min((B_{K^*} + S_{K^*})/2, S_{K^*+1})$. Whereas, for the $K^*$-th buyer the bid is close to $\max((B_{K^*} + S_{K^*})/2, B_{K^*+1})$. We can derive the long term average price, and surplus and deficit for the agents similarly. We leave out the exact expressions due to space limitations.

We acknowledge that the study of incentive compatibility under the notions of equilibrium in sequential games with incomplete information – where an agent can strategize thinking about her long term consequences (c.f. [37, 27]) in presence of learning is out of scope for this paper. This is similar to other contemporary works on learning in repeated games [29, 39, 30, 7].

# D  Synthetic Experiments

In this section, we present some additional synthetic experiments to study the impact of different parameters on the performance of our algorithm. Our methodology is same as mentioned in Section 6. Recall, that the negative regret for participant buyers/sellers is expected, as the regret increases as $(p(t) - p^*)$ for sellers, and $(p^* - p(t))$ for buyers.

## D.1  Example Systems:

In Figure 3, we have a $8 \times 8$ system with 5 matches. The non-participant regret, for both buyers and sellers, converges and assumes the $log(T)$. The participant regret, for both buyers and sellers, has more noise and the envelope grows as $O(\sqrt{T})$. Note that the regret that comes from price difference has opposite sign for buyers and sellers in each sample path. Hence, if regret plot of buyers is increasing with $T$ then it will decrease for sellers, and vice versa. We defer the simulation studies of other systems to appendix. We see this is Figure 4 where we have a $15 \times 15$ system with 10 matches simulated for $T = 100k$. The rest of the behavior in Figure 4 is similar to Figure 3.

## D.2  Impact of the gap $\Delta$

We first study the impact of changing the gap $\Delta$ on the performance, for a system of size $8x8$ and $K^* = 5$. In Figure 5, we study three gaps, $\Delta \in \{0.1, 0.15, 0.2\}$. Here, we observe that the convergence in the number of times an agent participates is delayed as we decrease the gap. As a result, there is an increase in the regret of non-participants, and social regret which is dominated by $\Delta$, whereas the participant regret is not directly impacted by $\Delta$ (after the initial stage) as it is dominated by the $O(\sqrt{T \log(T)})$ term.

---

[8]In this context, *close* means $\epsilon$ larger bids for buyers, and $\epsilon$ smaller bids for sellers for $\epsilon \gtrapprox 0$.

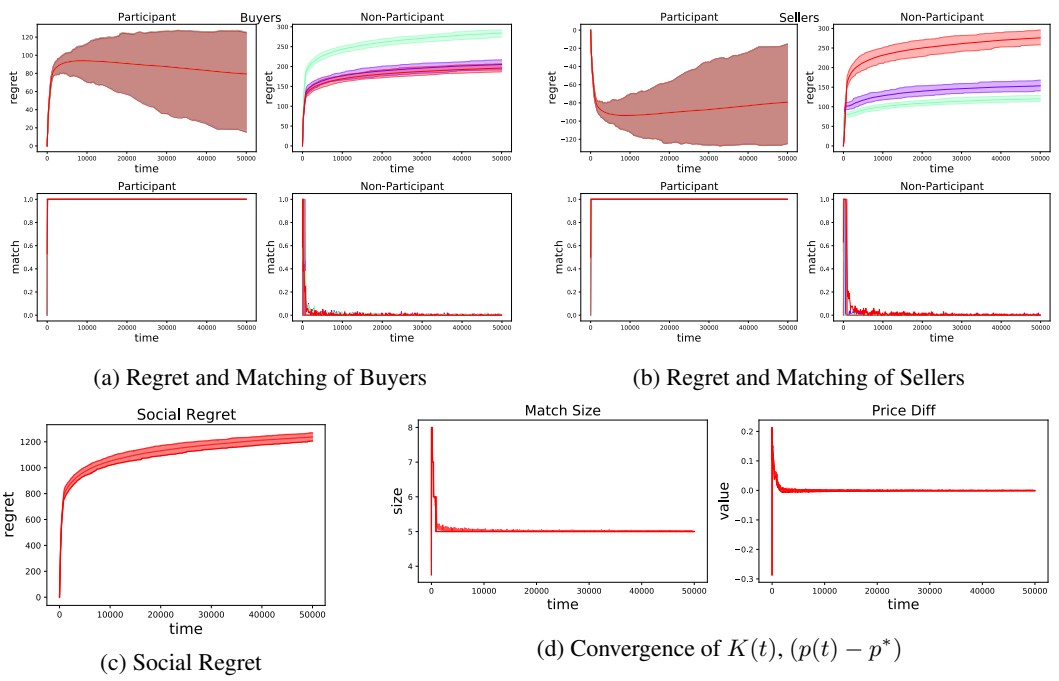

(a) Regret and Matching of Buyers

(b) Regret and Matching of Sellers

(c) Social Regret

(d) Convergence of $K(t), (p(t) - p^*)$

Figure 3: Double Auction $N = 8$, $M = 8$, $K^* = 5$, $\Delta = 0.2$, $\alpha_1 = 4$, and $\alpha_2 = 8$

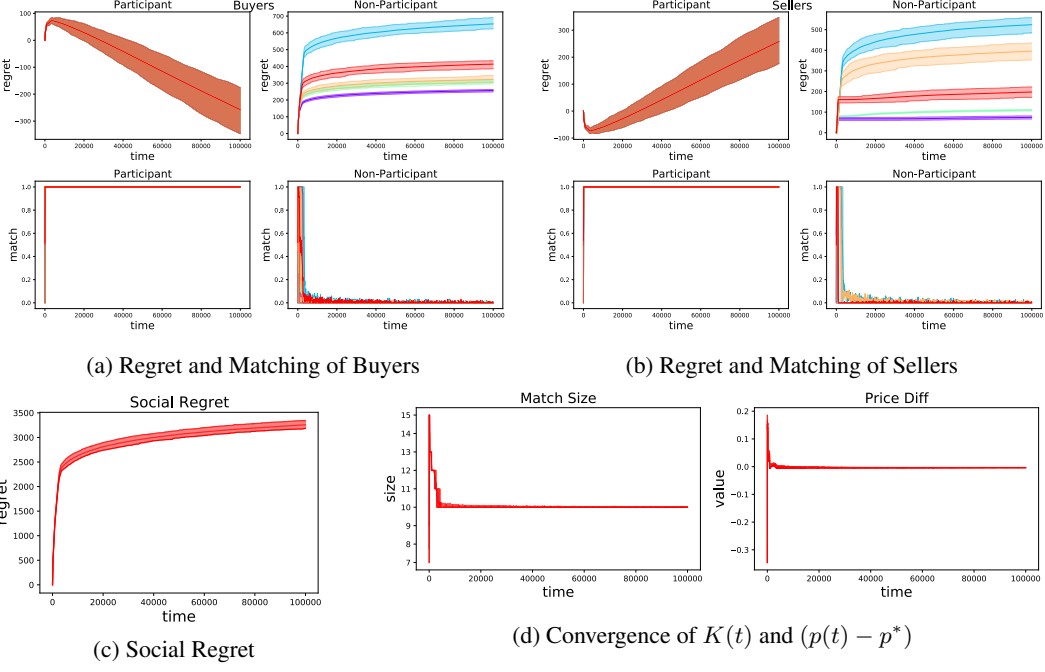

(a) Regret and Matching of Buyers

(b) Regret and Matching of Sellers

(c) Social Regret

(d) Convergence of $K(t)$ and $(p(t) - p^*)$

Figure 4: Double Auction $N = 15$, $M = 15$, $K^* = 10$, $\Delta = 0.4$, $\alpha_1 = 4$, and $\alpha_2 = 8$

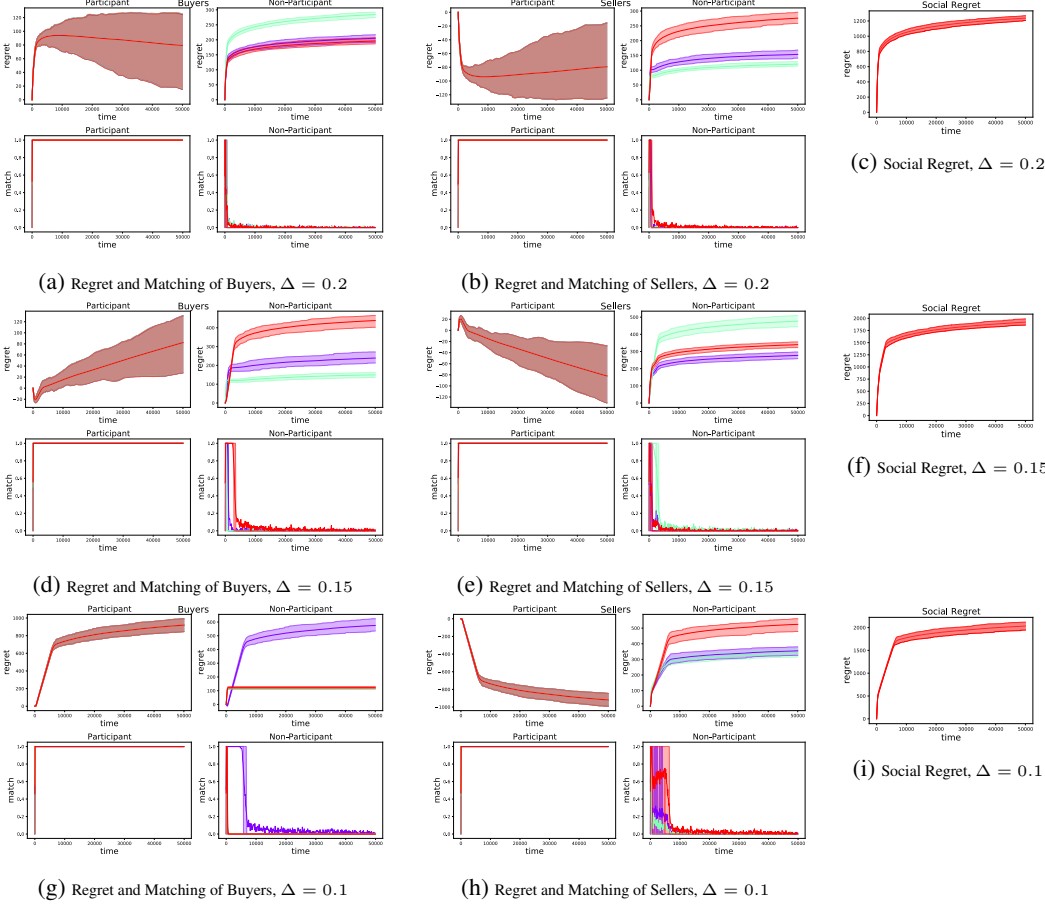

(a) Regret and Matching of Buyers, $\Delta = 0.2$

(b) Regret and Matching of Sellers, $\Delta = 0.2$

(c) Social Regret, $\Delta = 0.2$

(d) Regret and Matching of Buyers, $\Delta = 0.15$

(e) Regret and Matching of Sellers, $\Delta = 0.15$

(f) Social Regret, $\Delta = 0.15$

(g) Regret and Matching of Buyers, $\Delta = 0.1$

(h) Regret and Matching of Sellers, $\Delta = 0.1$

(i) Social Regret, $\Delta = 0.1$

Figure 5: Double Auction with varying $\Delta$ ($N = M = 8$, $K^* = 5$, $\alpha_1 = 4$, and $\alpha_2 = 8$).

### D.3 Impact of the sizes $M$, $N$, and $K^*$

We now study the impact of the size of the system, and number of true participants. First, with $M = N$, we vary $M$ while keeping the $(M - K^*)$ fixed. We observe in Figure 6 that the regret of the agents do not vary a lot, which is as expected from the theory. Next, we keep $M = N = 8$ fixed, while varying the participant size $K^*$. We see as $(M - K^*)$ increases, the regret increases in Figure 7 as suggested by the theory.

### D.4 Impact of size difference in $M$ and $N$.

In this part, we assess the size difference between number of sellers $M$, and number of buyers $N$. As the buyer and seller are symmetric, we keep $N = 8$, and $K^* = 5$ fixed. We next vary $M \in \{5, 8, 15\}$ to study the effect. In Figure 8, we observe the regret of non-participant buyers and sellers increases with increase in $M$. This is mainly because, the presence of more non-participant sellers is allowing the non-participant buyers to match with these non-participant sellers more increasing the regret. Similar observation holds for the social regret. The participant regret which is mostly dominated by price estimation error is mostly unaffected by this.

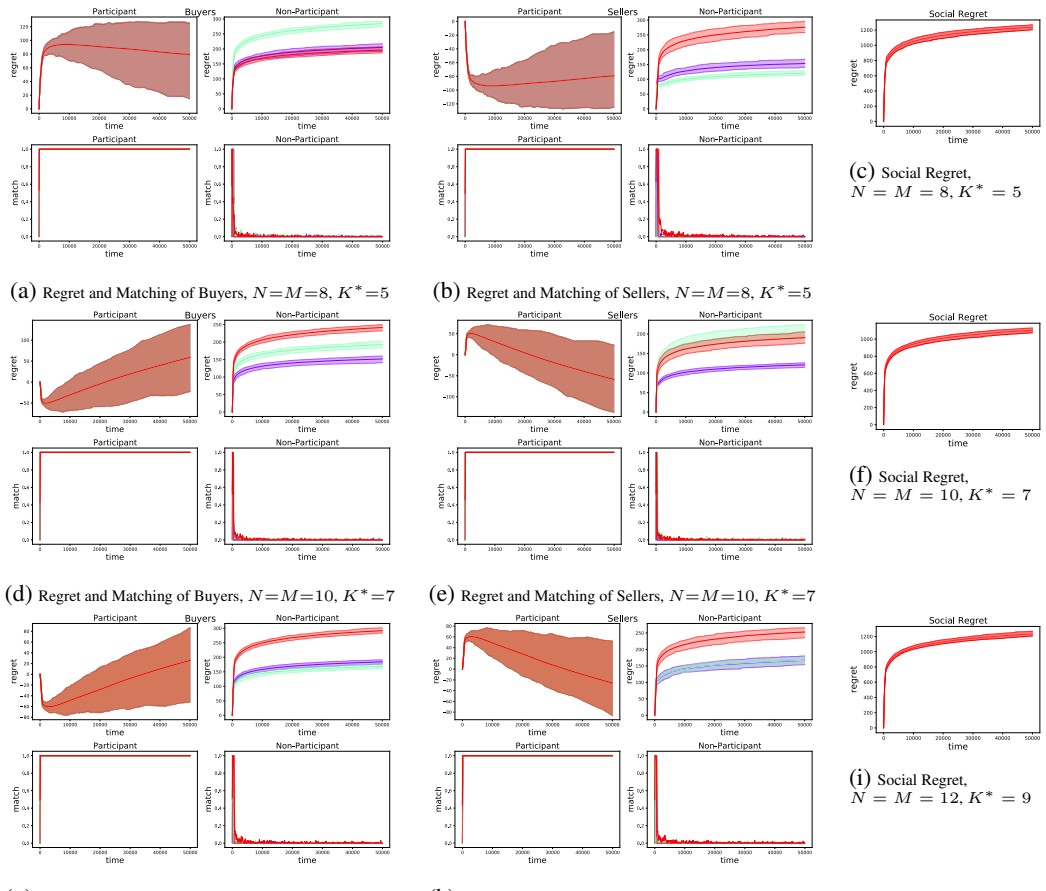

(a) Regret and Matching of Buyers, $N=M=8$, $K^*=5$

(b) Regret and Matching of Sellers, $N=M=8$, $K^*=5$

(c) Social Regret, $N = M = 8, K^* = 5$

(d) Regret and Matching of Buyers, $N=M=10$, $K^*=7$

(e) Regret and Matching of Sellers, $N=M=10$, $K^*=7$

(f) Social Regret, $N = M = 10, K^* = 7$

(g) Regret and Matching of Buyers, $N=M=12$, $K^*=9$

(h) Regret and Matching of Sellers, $N=M=12$, $K^*=9$

(i) Social Regret, $N = M = 12, K^* = 9$

Figure 6: Double Auction with varying $N = M$, and fixed $(M - K^*) = 3$ ($\Delta = 0.2$, $\alpha_1 = 4$, $\alpha_2 = 8$)

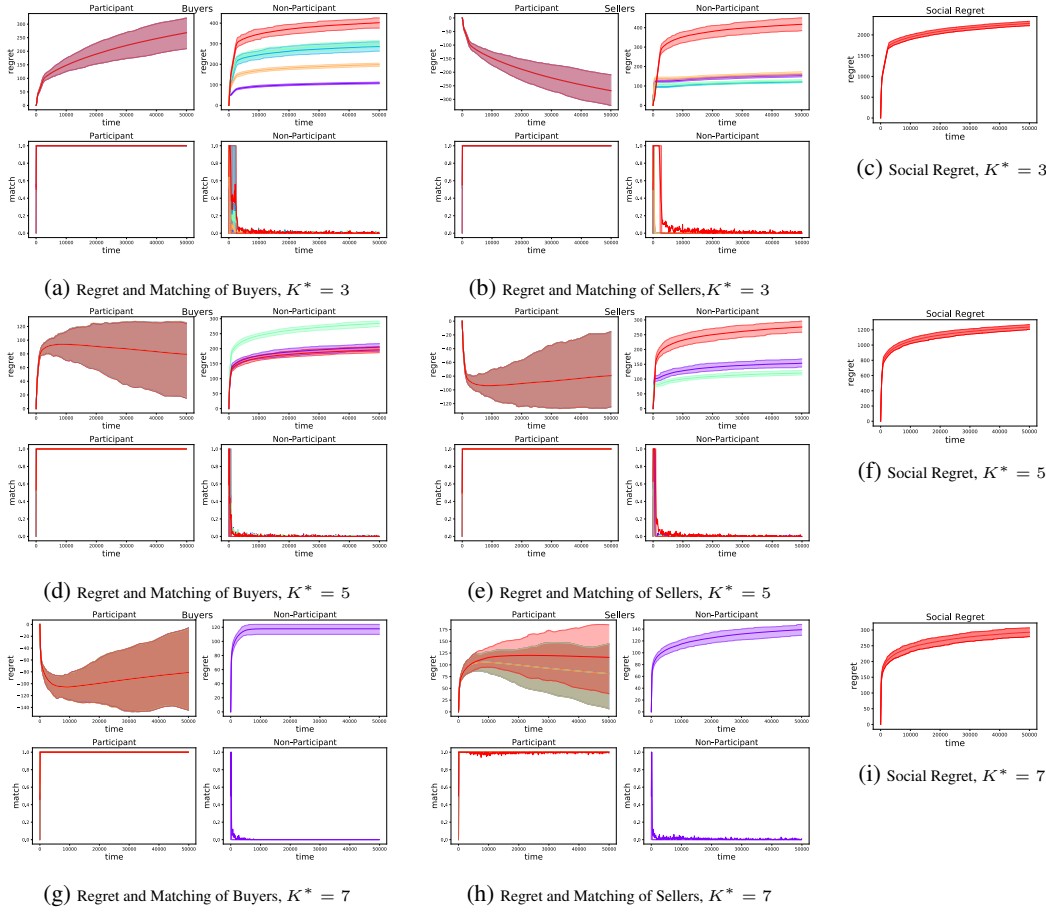

Figure 7: Double Auction with varying $K^*$ ($N = M = 8$, $\Delta = 0.2$, $\alpha_1 = 4$, $\alpha_2 = 8$)

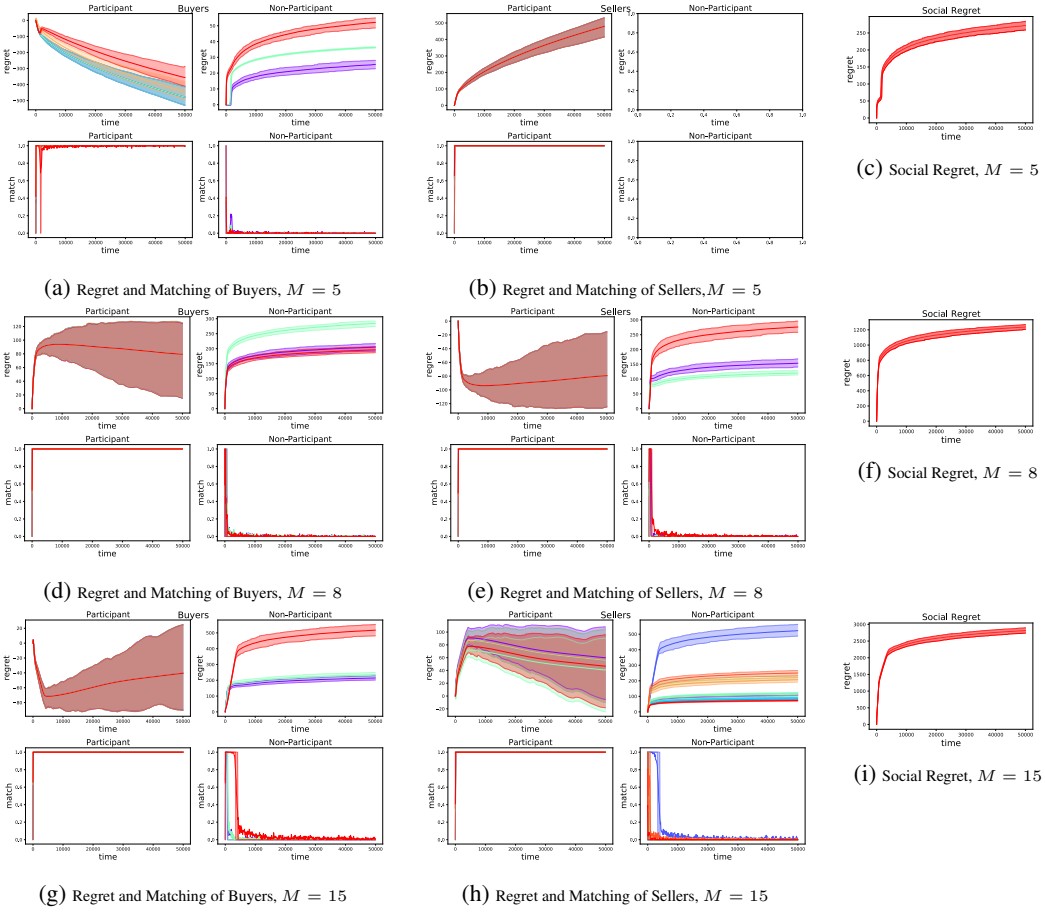

(a) Regret and Matching of Buyers, $M = 5$

(b) Regret and Matching of Sellers, $M = 5$

(c) Social Regret, $M = 5$

(d) Regret and Matching of Buyers, $M = 8$

(e) Regret and Matching of Sellers, $M = 8$

(f) Social Regret, $M = 8$

(g) Regret and Matching of Buyers, $M = 15$

(h) Regret and Matching of Sellers, $M = 15$

(i) Social Regret, $M = 15$

Figure 8: Double Auction with varying $M$ ($N = 8$, $K^* = 5$, $\Delta = 0.2$, $\alpha_1 = 4$, $\alpha_2 = 8$)