# OpenReview forum: "Double Auctions with Two-sided Bandit Feedback"
_NeurIPS.cc/2023/Conference — NeurIPS 2023 poster_

### Official Review · Reviewer_BEVv · 2023-06-26

**Soundness:** 3 good
**Presentation:** 3 good
**Contribution:** 3 good
**Rating:** 7
**Confidence:** 4

**Summary:**

This paper studies double auctions, where a set of $N$ buyers interacts with a set of $M$ sellers to trade some goods. This is a fundamental problem in economics that has been studied extensively. The specific mechanism that is studied in this paper is the average mechanism; it sorts the bids of the sellers and the buyers, identifies the best $k$ so that exactly $k$ trades can happen, and then posts as price the average of the bids of the kth seller and the kth buyer.

Formally, the authors study the following repeated setting: at each time $t$, each agent submits a bid, the average mechanism is implemented, and all the agents participating in a trade are revealed a noisy sample of their valuation. This paper aims to design a learning protocol that exhibits a sublinear regret with respect to the outcome of the average mechanism when all the agents declare their actual valuations. Note that in the model studied, the agents do not know their valuations but only learn it by participating in the auction and receiving bandit feedback.

The learning protocol proposed by the authors is simple: each agent maintains a (scaled) confidence interval of their valuation, the sellers bid according to LCB, and the buyers to UCB. This protocol yields $O(\log T /\Delta)$ regret regarding social welfare, which is tight. From the agent's perspective, the protocol exhibits an $O(\sqrt T)$ regret for agents belonging to the optimal solution and an $O(\log T/\Delta) $ regret for the others. These results are shown to be tight up to poly-log T terms.


**Strengths:**

- The study of economic problems from an online learning perspective is an active and fruitful line of research, and many works have appeared at NeuriPS and ICML
- The regret results are tight, and the algorithm is natural and elegant.
- The fact that the authors studied social welfare (i.e., total happiness) and individual utility (i.e., individual happiness) offers a compelling argument for implementing this mechanism in real life.
- Given the space constraints, the authors did a good job in presenting and motivating the problem and highlighting the crucial steps of the analysis


**Weaknesses:**

Minor comments:
- I do not find that surprising the fact that sellers want to bid according to their LCB
- Please spend some extra words on the definition of social welfare used in the paper. In the economic literature, social welfare is defined as the sum of the agents' utilities at the end of the trade. Thus, it also contains the sellers' valuations that retain their goods. The authors call social welfare the gain from trade, i.e., the variation in social welfare after and before the trade. As pointed out by [10], these two notions are equivalent when minimizing regret (due to its additive nature). Please clarify this point.
- It makes little sense to have an experimental paragraph in the main body and all the experimental results in the appendix. Either add some plots in the main body or defer everything to the appendix.
- The feedback in [10] is not bandit-like: they receive only ordinal information about the price posted and the agents' valuations. This is not enough to reconstruct the gain from trade (so it is strongly less informative than bandit feedback).


**Questions:**

None

**Limitations:**

See above

---

> ### Author Rebuttal · Authors · 2023-08-08
>
> **Social welfare definition:** We thank the reviewer for this pointer and will add the distinction from the classical definition of social welfare and what we consider. Further while defining social welfare, we will write a sentence crediting [10] for establishing the equivalence of the classical definition and what we consider while studying for regret minimization.
>
> **Weaker feedback in [10]:** We thank the reviewer for identifying this gap in our related work attribution. We will update the revision in Line 366 by adding a statement that –
>
> “*The work of [10] considers the single buyer and seller model under a much weaker ordinal feedback model, while in the present work we consider the multi-agent model under a stronger bandit feedback model. The feedback model in [10] is more restrictive since the gain from the trade cannot be estimated by the agents based on ordinal feedback while in our model, the gain from the trade can be estimated by each agent. However, our work considers the impact of multi-agent competition on regret minimization, which is not studied in [10].*”
>
> **Experimental plots in the main body:** We will follow the reviewer’s suggestion and use the extra page for the camera ready to move regret plots from the Appendix to the main body of the paper.

---

### Official Review · Reviewer_XUqU · 2023-07-03

**Soundness:** 4 excellent
**Presentation:** 4 excellent
**Contribution:** 3 good
**Rating:** 8
**Confidence:** 4

**Summary:**

This paper studies double auction markets where both buyers and sellers receive bandit feedback. There are $N$ buyers and $M$ sellers trading a single type of item in the market during a time horizon $T$. They don't know their own valuations so they need to learn through repeated interactions. The auctioneer implements the average price mechanism at each round. All buyers use UCB algorithms while all seller use LCB algorithms. The benchmark is defined to be the ideal market where true valuations are known and all players bid truthfully. The authors derive regret upper bounds for social welfare and individual utility. They also give regret lower bounds in the minimax sense.

**Strengths:**

1. To the best of my knowledge, this is the first paper that considers repeated double auctions with unknown valuations. Unlike in the standard repeated-auction setting, one need to deal with two-sided uncertainty in repeated double auctions. The regret analysis requires non-trivial techniques.

2. The analysis in the paper is limited to a specific variant of double auction mechanism, but the authors have discussed incentives and deviations, and have complemented the theoretical results with simulation study for completeness. I believe these results can inspire future work on a more general setting.

3. The paper is well written and organized. I particularly like the comments after the main results, which contribute to helping readers better perceive the regret bounds in theorems. The proof flow in the appendix is also very good.

**Weaknesses:**

1. The proof techniques may not be easily applied to more general settings, i.e., when agents don't follow UCB-type algorithms, when the auctioneer don't use the average price mechanism, or where true valuations are time-variant, sampled from some distribution.

**Questions:**

1. Line 148, $n_{s,j}(t)$ -> $n_{b,i}(t)$.

2. How to understand these regret bounds if there are ties. For instance, if $B_{K^*} = B_{K^*+1}$, will the regret on social welfare be arbitrarily bad? What if $\Delta=0$?

3. I personally suggest making the proof outline more concise, so that the main body of the paper can have space to put some results of simulation experiments.

4. Line 615. Does it mean $\beta\geq 2$ is enough for proving Regret=$O(\log T)$?

**Limitations:**

The authors acknowledge that the study of incentive compatibility is out of scope for this paper. However, they have given enough discussion on incentives in Appendix C.

---

> ### Author Rebuttal · Authors · 2023-08-08
>
> We thank the reviewer for their valuable comments, and highlighting insightful future directions. Please find our response below.
>
> **General Applicability of Proof Techniques:**  In bandit literature, handling stochastic reward and time-variant reward requires different algorithmic and technical ideas. We acknowledge this, and leave handling time-variant ture valuation open. We will add this to our future work discussions.
>
> Our proof techniques of Social regret relies only on convergence to the correct set of buyers and sellers, and can be adapted to other mechanisms with a reasonable effort. However, our techniques require substantive modifications to handle individual regret in other mechanisms as price discovery changes drastically with mechanism.  We leave studying general double auction mechanisms with bandit feedback as a future work (see line 389-390 in the paper).
>
> **Adding Plots in the Main Body:** Thank you for this suggestion. If accepted, we will utilize the additional page in the camera ready to add simulation plots in the main paper.
>
> **$\Delta = 0$ is not feasible:**  $\Delta = 0$ (where $\Delta$ is defined in line 182) implies $p^* = S_{K^*} = B_{K^*}$. However, Double auction only matches a buyer with bid higher than a  seller. So if $K^*$-th buyer and $K^*$-th seller participate we must have $S_{K^*} < B_{K^*}$. Therefore, we always have $\Delta > 0$.
>
> **Ties in Valuation:** $B_{K^*} = B_{K^*+1}$ is an interesting setting. Understanding this behavior formally is out of scope. We provide an informal discussion here.  In the case where $B_{K^*} = B_{K^*+1}$ and $S_{K^* + 1} > B_{K^*}$, i.e. there are $(K+1)$ buyers and $K$ sellers. A tie-breaking needs to be employed. We assume the tie-breaking happens randomly with equal probability.  Under bandit feedback the UCB of $(K^*+1)$-th and $K^*$-th buyer will both be higher than LCB bid from the $K^*$-th seller with high probability. As exactly one of the two buyers match every round the social regret will remain mostly unchanged orderwise in N, M, K and T.  The individual regret analysis is more complicated. We conjecture the number of times  $K^*$-th and $(K^* + 1)$-th buyer match satisfy,  $1/polylog(T) \leq  n_{K^*+1}(T) / n_{K^*}(T) \leq polylog(T)$. Otherwise, the less matched buyer will have a higher UCB whp. Thus due to fluctuations in the UCB of these two buyers they will incur $polylog(T)$ regret. However, this is orderwise negligible to the $\sqrt{T}$ individual regret from the price estimation error for these two buyers.  A formal treatment requires considering *all possible ties*, specifying *tie-breaking rules*, and utilizing arguments similar to the above to give regret guarantees. Therefore, we will leave studying *'Ties in valuation'* open, and add it to our *Conclusion and Future Work* section.
>
>
> **$\beta > 2$ suffices:** Thank you for pointing this out. Yes $\beta > 2$ suffices for our regret upper bounds, with increased constant term. E.g. Theorem 1 will hold with $MNb_{max} \zeta(\beta / 2)$ instead of $MNb_{max} \pi^2/6$, where $\zeta(x)$ is the Riemann zeta function which is finite for $x > 1$.  We will update this in the final version.

---

> > ### Comment · Reviewer_XUqU · 2023-08-17
> >
> > Thanks for the rebuttal. I don't have any further questions.

---

### Official Review · Reviewer_bD4t · 2023-07-04

**Soundness:** 3 good
**Presentation:** 3 good
**Contribution:** 2 fair
**Rating:** 5
**Confidence:** 3

**Summary:**

The paper studies an online learning  problem related to double auctions. In particular, the paper assumes that the auctioneer uses a average price mechanism. The goal is to study the online learning problem faced by sellers and buyers that do not know they own valuations. The authors design a algorithm based on upper confidence bounds that if applied by all the sellers and buyers guarantees low regret both for each individual and socially. In particular, they show that the social regret is upperbounded by $O(log T/\Delta)$, where $\Delta$ is the minimum price gap, while the individual regret is at most $O(\sqrt T)$.

**Strengths:**

The setting is interesting and of practical relevance. The proposed algorithm is easy to implement and guarantees good regret bounds.

**Weaknesses:**

The algorithmic approach is straghtforward and does not introduce substantial new ideas.

I am confused by your choice to use instance dependent regret bound in some cases (depending on $\Delta$) and some instance independent regret bounds $\sqrt T$. I don’t see any conceptual difference between the case in which you apply instance dependent and instance independent bound. For instance, your Lemma 21 (that proves the instance dependent bound) employs arbitrary small difference between the buyer and seller valuations. Hence, it does not rule out logarithm instance dependent bound depending on $\Delta$.

**Questions:**

Please clarify why your are using both instance dependent and instance independent regret bounds. Why do you handle individual regret and social regret in different ways?

**Limitations:**

Yes

---

> ### Author Rebuttal · Authors · 2023-08-08
>
> **1. Regarding instance dependent versus instance independent bounds**
>
> ***All our upper bounds in Theorems 1 and 2 are instance dependent***, including the $\sqrt{T}$ for participating agents. Theorem 1 bounds the social-welfare regret as a function of $\Delta$ and is thus instance dependent. In theorem 2, the regret upper bounds for both the participating buyers(sellers) and the non-participating buyers(sellers) depend on instance dependent terms such as $K^*$ and $\Delta$. Observe the $\Delta$ dependence in the second order $\log(T)$ term for the participating buyers and sellers’ regret. Thus, our regret upper bounds are instance dependent bounds.
>
> *We propose to add a statement and highlight that all our upper bounds are instance dependent in the revised version.*
>
>
>
> ***$\sqrt{T}$ Instance dependent Individual Regret:*** Among the instance dependent upper bounds, the regret for the participating buyers and sellers is the non-standard term of $\sqrt{T}$. This is unlike $\log(T)$ instance dependent bounds in typical multi-armed-bandits. At a high-level, a $\sqrt{T}$ term for the regret shows up for participating agents because of having to do *price-discovery* which is a continuous variable. To solidify this intuition, consider estimating the mean of a continuous random variable using $t$ i.i.d. Samples. It is known (for ex. Using Central Limit Theorem arguments)  that the estimated mean will be about $\mathcal{O}\left(\frac{1}{\sqrt{t}}\right)$ away from the true mean. In the price-discovery setting, this error in estimating the true valuation will accumulate over time to a $\sum_{t=1}^T\mathcal{O} \left(\frac{1}{\sqrt{t}}\right) = \mathcal{O} \left(\sqrt{T}\right)$ regret term.  Moreover, the system needs to eliminate suboptimal *matching* by playing them $O(log(T))$ times. This reflects in the $O(log(T))$ components of the individual regrets.
>
> ***$O(\log(T))$ Instance dependent Social Regret:*** In the social welfare regret, the *price discovery* does not play any role, because in each round the prices cancel out in the social welfare. Only sub-optimal *matching* used in each round contributes to the regret leading to decomposition in line 258. $O(log(T))$ number of play of each of the finite sub-optimal matchings suffices to separate the optimal matching, creating $O(log(T))$ social welfare regret.
>
> ***Our lower bound on social welfare regret is instance dependent:*** The lower bound given in Appendix B.7 on the social welfare is instance dependent and order-wise captures the upper bound. We are able to give instance dependent lower bounds by establishing a coupling between our system and a combinatorial semi-bandit system in Proposition 22.
>
> ***Our Lower bound for individual regret (Lemma 21) is instance independent:*** As the reviewer correctly notices, our lower bound for price-discovery in Lemma 21 is instance independent. Note we already mention this as a *minimax bound*. We do not have an instance dependent lower bound for price discovery in our work. Establishing an instance dependent lower bound is a challenging open problem as it requires us to formalize the mean-estimation intuition to obtain a $\sqrt{T}$ type instance dependent lower bound. Doing so requires circumventing several technical challenges such as estimation from adaptively collected samples through a bandit policy. The correlations among the samples are  further complicated by the impact of market competition where a sample for the valuation is received only when an agent is matched which in turn is dependent on the other agents’ perceived valuation. These technical challenges render an instance dependent lower bound for the pricing beyond the scope of the present work.
>
> *We propose to make this explanation clearer in the revision and will highlight that the only instance independent regret bound is the individual regret. We will also pose as an open problem/future work of obtaining a $\sqrt{T}$ type instance dependent lower bound on individual regret.*
>
>
> **2. Regarding algorithmic simplicity and technical contributions**
>
> ***We view the simplicity of our algorithm as a strength:*** We note that our algorithm (sellers bid LCB, and buyers bid UCB) attains optimal social regret, and sub-linear individual regrets. Our key contribution is to show under the presence of two-sided uncertainty of double auction markets, simple UCB-LCB based algorithm can succeed. The fact that a simple algorithm can attain this performance is a strength and contribution of the paper. Thus, we respectfully push-back on the opinion that simplicity is a weakness.
>
> ***Technical Challenges and contributions in the analysis:*** We want to remark that despite the algorithm being simple, its analysis is novel and does not follow existing works in multi-armed bandits. The key challenge in the analysis is that the information across agents are different resulting in heterogeneity, as we explain in Section 4.1. Thus, even though the algorithm appears simple, *new ideas are needed for the multi-agent analysis.*

---

> > ### Comment · Reviewer_bD4t · 2023-08-21
> >
> > Thanks for the detailed response that clarifies most of my doubts. I've update my score accordingly.
> >
> > I encourage the author to better highlight that the regret for the participants depends also on $\Delta$. For instance, I find Table 1 misleading. The dependence from $\Delta$ should appear also in the $\sqrt T$ regret bound.
> >
> > Lemma 21 is not very meaningful in the context of you work. If I understand correctly, your upper bound on the class of instances used in the proof is arbitrary large! Again, Table 1 suggests that you have an almost matching lower bound for participants but this is not the case.

---

> > > ### Author Response · Authors · 2023-08-21
> > >
> > > We thank the reviewer for a through and constructive feedback that is helping our draft!
> > >
> > > We will add the discussion from this rebuttal and in particular explicitly include $\Delta$ into Table 1. We will also highlight the limitation of our results in Table 1 that for individual regret,  our upper bound is instance dependent while the lower bound (Lemma 21) is instance independent. In the conclusion section, we will list  deriving instance dependent lower bounds for individual regret as an open problem.

---

### Official Review · Reviewer_51dr · 2023-07-06

**Soundness:** 3 good
**Presentation:** 2 fair
**Contribution:** 3 good
**Rating:** 5
**Confidence:** 3

**Summary:**

This work considers learning in a two-sided double auction setting in which both sides must confront uncertainty over their valuations (which are realized upon winning in the auction) and choose to adhere to the same protocol in order to perform that learning. The work shows that buyers and sellers bidding less aggressively - e.g., buyers bidding higher than they otherwise might, and sellers bidding lower than they otherwise might - leads to faster learning than the usual alternative (Optimism in the Face of Uncertainty). One note is that strategic behavior and any Nash equilibrium will be a change in the other direction, and involve bidders bidding more aggressively (buyers lower / sellers higher).

Using this strategy, the paper develops bounds on social welfare and individual participant regret.

It is mentioned in the appendix that individuals may have incentives to deviate, but in the context of robustness of results, strategic behavior is not considered.

**Strengths:**

The work makes an interesting connection between double auctions and bandit learning, and would seem to be the first to tackle learning on both sides of the market.

**Weaknesses:**

The recommended strategy is quite simple - bidding higher for buyers or lower for sellers, which results in more trade happening than e.g. bidding at expectation or bidding more aggressively, and it is this encouragement of trade that drives results. The authors note that if both sides bid too aggressively then it can slow down trade and hence slow down learning. Incentives from best responding however will push buyers and sellers in the other direction.

The protocol model assumes coordination between buyers and sellers on the algorithm to use in order to bid, which could in a real world setting encourage collusion among participants and/or platform disintermediation.

**Questions:**

Double auctions are ubiquitous as mentioned in the discussion. Which many-to-many double auction bandit settings do you view as the right motivation for this work?

**Limitations:**

No - discussions of collusion or platform disintermediation would be beneficial for the work.

---

> ### Author Rebuttal · Authors · 2023-08-09
>
> We thank the reviewer for the comments.
>
> **Which many-to-many double auction bandit settings is the motivation?**
> Online learning in economic markets is an active area of research evident from the abundance of research works in the past few years. This is also acknowledged by *Reviewer BEVv*. Our work contributes to this area by initiating the study of double auction.
> We draw motivation from multiple practical applications, e-commerce, bidding in wireless spectrum, cloud computing markets to name a few. Please see line 38 - line 47 for details.  Our objective here is not to model one particular application in detail, but to develop a framework to study repeated double auctions under bandit learning.
>
> As a stylized example consider a specific task, say *image labeling*, in a *decentralized crowdsourcing marketplace*. There are multiple *labelers (sell-side)*, each having her own valuation of labeling a batch of images determined by the time taken to label. Here time taken for labeling is unknown, stochastic, and different for each labeler. The labeling task is accomplished with comparable accuracy by each labeler. There are multiple *companies (buy-side)* that want to get a batch of images labeled in each interaction. Each company has its own valuation of a batch of labeled images at the accuracy of the pool of labeler under consideration. However, the accuracy and hence the valuation of each company is a-priori unknown. The companies and labelers can transact -- labelers are paid by the companies for a batch of labeled images -- through a repeated double auction market. The accuracy of labeled images by a company, and time taken for labeling by a labeler can be learned only through *image-labeling (bandit feedback)*. Our work captures this scenario.
>
>
> **We view the simplicity of our algorithm as a strength:** We note that our algorithm (sellers bid LCB, and buyers bid UCB) attains optimal social regret, and sub-linear individual regrets. Our key contribution is to show under the presence of two-sided uncertainty of double auction markets, simple UCB-LCB based algorithm can succeed. The fact that a simple algorithm can attain this performance is a strength and contribution of the paper. Thus, we respectfully push-back on the opinion that simplicity is a weakness.
>
> **Incentives from best responding however will push buyers and sellers in the other direction:**  We note that indiscriminately reducing the bids for a buyer, and increasing the bid for a seller is not always best. Indeed, too low a bid from buyer or too high a bid from seller can leave them unmatched leading to high regret. Note that for the true-participant sellers and buyers who benefit from trade the UCB and LCB decays roughly as $O(1/\sqrt{t})$ at time $t$. So for these buyers and sellers, the bids converge to their true valuation. For true-non-participating buyers and sellers there is no gain from trade. UCB-LCB based bidding happens in the vicinity of true valuation while ensuring their participation in trade is vanishing (only $O(\log(T))$ in $T$ rounds). Therefore, UCB-LCB bidding is as incentivized as bidding their respective true valuation for all agents (quantified by the individual regrets). We discuss the incentives of deviation from true valuation in detail in Appendix C along with a summary in our main paper (see line 212 - 221).
>
>
> **Collusion among participants and/or Platform disintermediation:** Thank you for bringing up the possibility of collusion and platform disintermediation within the protocol model. We acknowledge these drawbacks of protocol model, and will mention them in our revised *Conclusion and Future Work* section. However, going beyond the protocol model where the agents are taking part in a repeated game with unknown valuation is out of scope for this paper. In fact, to the best of our knowledge,  this limitation is shared by almost all works on Bandit learning in Markets with stochastic rewards as these works also adopt a protocol model (see line 25 - line 29).

---

> > ### Comment · Reviewer_51dr · 2023-08-21
> >
> > Thank you for the response - based on this and other reviewers, my initial take on the setting was too harsh. I would like to see more general results but in current form it is of interest for the NeurIPS audience.

---

> > > ### Author Response · Authors · 2023-08-21
> > >
> > > We thank the reviewer for their thorough and constructive feedback through this process!  We will add the discussion from this rebuttal including open problems/limitations of the current results and future directions into the revised version.

---

### Decision · Program_Chairs · 2023-09-21

**Decision:**

Accept (poster)

**Comment:**

This paper looks at a double auction where multiple buyers and sellers both compete; the paper builds on the two-sided bandit feedback setting that has been fleshed out over the last few years in the NeurIPS/ICML/COLT communities, where buyers/sellers learn about their valuation in a round after bidding and receiving their outcome.  Reviewers appreciated the novelty of the setting (this AC also believes this is the first paper considering a double auction + unknown vals), the breadth of results, and the exposition/presentation of the work.  We’d ask the authors to pay special attention to their comprehension rebuttals and reviewers’ discussion responses; reviewers brought up some great points that would make this paper much stronger should it appear at NeurIPS or elsewhere.